# Metabolic gene regulation by *Drosophila* GATA transcription factor Grain

**Krista Kokki**[1,2☯], **Nicole Lamichane**[1,2☯], **Anni I. Nieminen**[1,2], **Hanna Ruhanen**[1,3], **Jack Morikka**[1,2], **Marius Robciuc**[1,2], **Bohdana M. Rovenko**[1,2], **Essi Havula**[4], **Reijo Käkelä**[1,3], **Ville Hietakangas**[1,2]*

**1** Molecular and Integrative Biosciences Research Programme, Faculty of Biological and Environmental Sciences, University of Helsinki, Helsinki, Finland, **2** Institute of Biotechnology, University of Helsinki, Helsinki, Finland, **3** Helsinki University Lipidomics Unit (HiLIPID), Helsinki Institute for Life Science (HiLIFE) and Biocenter Finland, Helsinki, Finland, **4** Stem Cells and Metabolism Research Program, Faculty of Medicine, University of Helsinki, Helsinki, Finland

☯ These authors contributed equally to this work.
* ville.hietakangas@helsinki.fi

**Data Availability Statement:** The RNAseq data is available in GEO, with the accession number GSE166681. https://www.ncbi.nlm.nih.gov/geo/query/acc.cgi?acc=GSE166681.

## Abstract

Nutrient-dependent gene regulation critically contributes to homeostatic control of animal physiology in changing nutrient landscape. In *Drosophila*, dietary sugars activate transcription factors (TFs), such as Mondo-Mlx, Sugarbabe and Cabut, which control metabolic gene expression to mediate physiological adaptation to high sugar diet. TFs that correspondingly control sugar responsive metabolic genes under conditions of low dietary sugar remain, however, poorly understood. Here we identify a role for *Drosophila* GATA TF Grain in metabolic gene regulation under both low and high sugar conditions. *De novo* motif prediction uncovered a significant over-representation of GATA-like motifs on the promoters of sugar-activated genes in *Drosophila* larvae, which are regulated by Grain, the fly ortholog of GATA1/2/3 subfamily. *grain* expression is activated by sugar in Mondo-Mlx-dependent manner and it contributes to sugar-responsive gene expression in the fat body. On the other hand, *grain* displays strong constitutive expression in the anterior midgut, where it drives lipogenic gene expression also under low sugar conditions. Consistently with these differential tissue-specific roles, Grain deficient larvae display delayed development on high sugar diet, while showing deregulated central carbon and lipid metabolism primarily on low sugar diet. Collectively, our study provides evidence for the role of a metazoan GATA transcription factor in nutrient-responsive metabolic gene regulation *in vivo*.

## Introduction

Animal physiology is constantly readjusted in changing nutrient landscape. The ability to control systemic metabolism with respect to nutrient content is due to so-called nutrient sensing pathways. They are composed of inter- and intracellular signaling pathways and gene regulatory networks that control the activity of metabolic genes and gene products [1,2]. A key regulator of dietary sugar responsive gene expression is the heterodimeric basic helix-loop-helix

**Funding:** Funding was provided by: Academy of Finland (MetaStem Center of Excellence funding 312439) to VH, Sigrid Juselius Foundation to VH, Novo Nordisk Foundation (NNF18OC0034406 and NNF19OC0057478) to VH, Jane and Aatos Erkko Foundation to VH, and Integrative Life Science Doctoral Program to NL. The funders did not play any role in the study design, data collection and analysis, decision to publish, or preparation of the manuscript.

**Competing interests:** The authors have declared that no competing interests exist.

transcription factor Mondo/ChREBP-Mlx [3]. Mondo-Mlx is activated by glucose-6-phosphate and other phosphorylated hexoses and it controls metabolic genes, such as those involved in glycolysis, pentose phosphate pathway (PPP) and lipogenesis, through carbohydrate response element (ChoRE) motifs [4–7]. In *Drosophila*, compromised activation of Mondo-Mlx leads to sugar intolerance, *i.e.* inability to survive on sugar-containing diets, which is associated with defects in clearing glucose from circulation as well as impaired $NADP^+$ reduction through PPP [5,8,9]. Sugar intolerance is also observed in the dietary specialist *Drosophila* species, *Drosophila sechellia*, which is adapted to low sugar diet and displays weaker sugar-induced activation of many Mondo-Mlx target genes [10,11].

In addition to metabolic genes, *Drosophila* Mondo-Mlx regulates genes encoding transcription factors (TFs), such as Klf-10 ortholog *cabut* and Glis2 ortholog *sugarbabe*, both of which are also necessary for larval sugar tolerance [5,6,12]. *sugarbabe* gene is regulated through ChoREs located in its promoter, and it promotes the expression of metabolic genes, such as those involved in fatty acid biosynthesis [6,13]. In addition to TF targets, Mondo-Mlx controls hormonal signaling pathways, such as an Activin encoding gene *dawdle*. Mondo-Mlx promotes the expression of *dawdle*, which contributes to metabolic adaptation to high sugar diet by activating signaling through the Activin receptor Baboon and transcription factor Smad on X (Smox) [6,14,15]. The Dawdle/Smox signaling converges with Sugarbabe regulation, as activation of *sugarbabe* expression in response to high sugar diet is compromised in animals depleted of Dawdle or Smox [6]. One of the metabolic consequences of defective signaling by Dawdle is the accumulation of TCA cycle intermediates and consequent hemolymph acidosis [14]. Despite these advances, our understanding on sugar-responsive gene regulators and their targets remains incomplete.

GATA binding zinc finger transcription factors are conserved among eukaryotes and have well-established roles during animal development [16]. The six vertebrate GATA factors have been divided into two subfamilies, hematopoietic GATA1/2/3 and cardiac GATA4/5/6 clades. However, their expression and physiological roles extend beyond these tissues [16]. *Drosophila* genome includes five GATA encoding genes, of which *grain* (*grn*) is orthologous to GATA1/2/3, whereas *serpent* (*srp*), *pannier* (*pnr*), *GATAd* and *GATAe* belong to the GATA4/5/6 clade [17]. Grain is involved in morphogenetic processes, contributing to the shaping of adult legs and the larval posterior spiracles [18], as well as neuronal cell fate specification [19] and axon guidance [20]. GATA transcription factors have also been reported to have metabolism-related roles. Mammalian GATA-2 and GATA-3 are expressed in white adipocyte precursors and their downregulation is necessary for terminal adipocyte differentiation [21]. Reduced adipose expression of GATA-2 and GATA-3 is observed in mouse models of obesity [21]. Mechanistically, GATA-2 and GATA-3 repress the expression of Peroxisome proliferator-activated receptor (PPAR) gamma and directly interact with CCAAT/enhancer binding proteins alpha and beta [22]. In yeast, GATA transcription factors have well-established functions in nitrogen metabolism, downstream of the nutrient sensing protein kinase TOR complex 1 [23,24]. Moreover, evidence from human pathogenic fungi *Blastomyces dermatitidis* has established a role for GATA transcription factor SREB in biosynthesis of triacylglycerol (TAG) and ergosterol, as well as lipid droplet formation [25].

Through an unbiased *in silico* search of candidate transcription factors involved in sugar-dependent gene regulation, we have discovered that *Drosophila* GATA transcription factor Grain contributes to the regulation of a specific subset of sugar responsive genes. Grain is expressed in several metabolically relevant tissues, displaying high constitutive expression in the midgut and sugar-inducible expression in the fat body. Grain activity is necessary for lipogenic gene expression in the anterior midgut. Consistent with the sugar-dependent and -independent tissue specific roles, Grain contributes to the regulation of central carbon and lipid

metabolism primarily on low sugar diet, while maintaining larval growth on high sugar conditions. Moreover, Grain shares common targets and genomic binding sites with Sugarbabe, including lipogenic genes *ATP citrate lyase* (*ATPCL*), *Acetyl-CoA synthetase* (*AcCoAS*), *Acetyl-CoA carboxylase* (*ACC*), and *Fatty acid synthase* (*FASN1*). While Sugarbabe promotes the lipogenic gene expression primarily on high sugar diet, Grain mainly contributes to their expression on low sugar diet. Thus, Grain integrates with the sugar-responsive gene regulatory network to control energy metabolism in *Drosophila* larvae.

## Results

### Predicted role for GATA transcription factor Grain in the regulation of sugar-responsive genes

In order to identify novel candidate transcription factors involved in sugar-responsive gene regulatory network, we re-analyzed our previously published RNA-seq dataset with sugar-regulated genes in 2nd instar *Drosophila melanogaster* larvae [6]. Gene expression profiles of whole larvae acutely (8 h) transferred on high sugar diet (HSD, 10% yeast + 15% sucrose) or maintained on low sugar diet (LSD, 10% yeast) were compared and differentially expressed genes were identified. In total, 1532 genes were found to be significantly (adj.p.val <0.05) upregulated, and 1043 genes were downregulated on HSD. These gene sets were used for *in silico* motif enrichment to identify candidate transcriptional regulators. DREME *de novo* motif discovery, separately for up- and downregulated gene sets, was followed by comparison to databases of known transcription factor binding motifs by TOMTOM (**S1 Table**). One of the highest enriched motifs from sugar upregulated gene set was BYGATAAG, which was predicted as a likely binding site for GATA transcription factors (**Fig 1A**). Since the GATA-factors share nearly similar motifs, we took a closer look at the *de novo* enrichment and identified the best matching motifs, which singled out Grain (Grn), the *Drosophila* GATA1/2/3 ortholog, as the candidate with the lowest E.value (**Figs 1B and S1A**).

The ENCODE database contains ChIP-seq data for two *Drosophila melanogaster* GATA TFs, Grain and GATAd, which allowed us to further analyze their possible involvement in the regulation of sugar responsive target genes. To this end, we re-analyzed the raw datasets (ENCSR909QHH and ENCSR245UCO) and tested their overlap with the sugar-responsive up- and downregulated gene sets (**Figs 1C and S1B–S1D**). The overlap between sugar-responsive genes and GATAd targets was less than 3% (**S1B and S1C Fig**), while the overlap between upregulated sugar-responsive genes and Grain targets was strikingly high, >50% (**Fig 1C**). In contrast, the overlap between Grain targets and the genes downregulated by sugar was lower (25%) (**S1D Fig**). The Grain ChIP-seq further validated our *in silico* target prediction, as the majority of GATA motif containing genes were found to be direct Grain targets (**Fig 1D**).

### Grain is necessary for sugar tolerance and growth

To examine the possible role of Grain in sugar-induced responses, we analyzed its expression upon sugar feeding in whole *Drosophila* larvae. Interestingly, *grain* gene expression was modestly upregulated upon 8 h on HSD. This upregulation was not detected in *mlx*[1] mutant larvae (**Fig 2A**), suggesting that *grain* expression is regulated by Mondo-Mlx. Previously identified transcription factors of the Mondo-Mlx–dependent gene regulatory network, including Mondo-Mlx itself, Sugarbabe and Cabut, are all necessary for growth on high sugar diet [5,6,12]. *grain* mutant alleles are embryonic lethal [18], but the use of ubiquitous (Tub-GAL4>) *grain* knockdown by RNAi allowed the analysis of larval and pupal loss-of-function phenotypes (denoted as *grn* in the Figures). On LSD, *grain* knockdown did not significantly

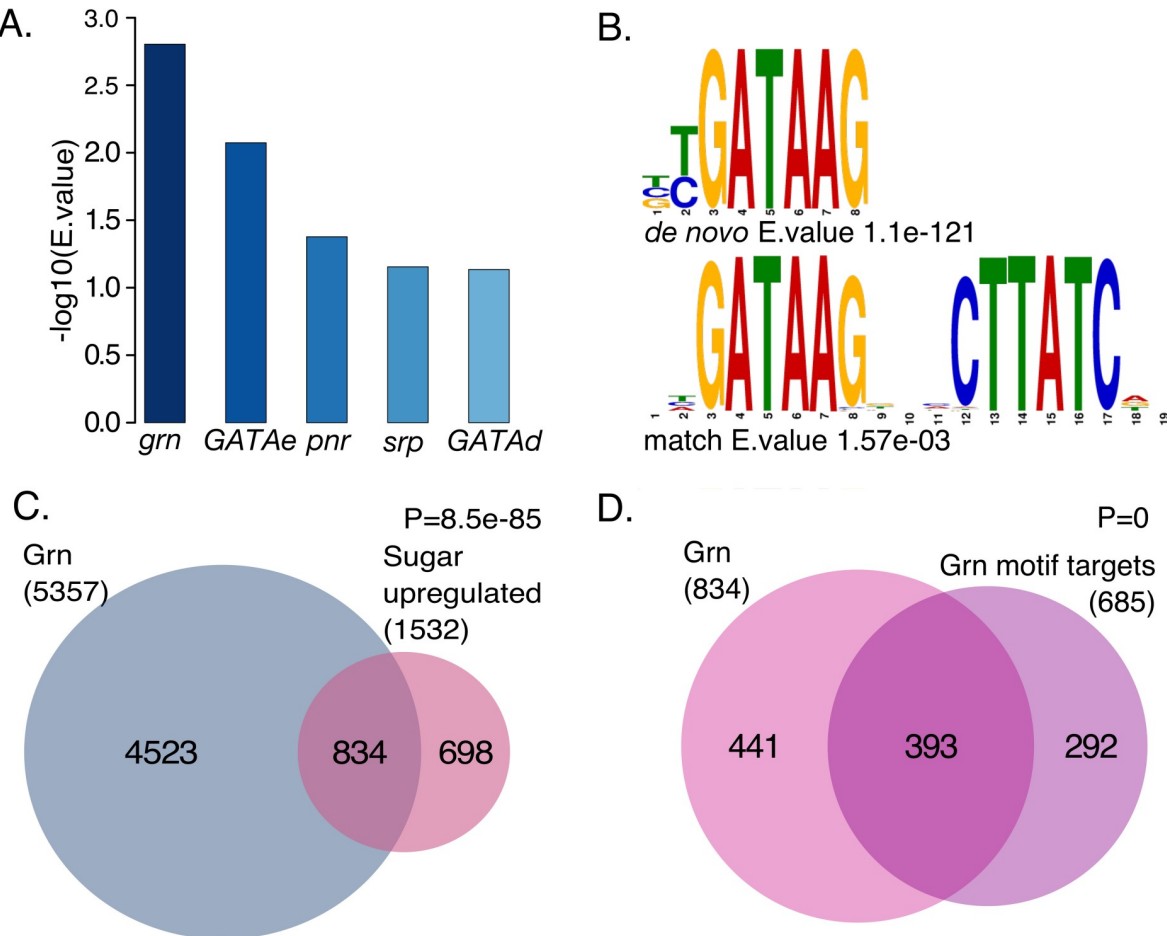

**Fig 1. *In silico* prediction of GATA TF Grain as a regulator of sugar responsive genes. A.** Visualization of E.values of GATA transcription factor motif matches (TOMTOM) to selected *de novo* predicted motifs (DREME) in sugar upregulated gene set (RNA-seq) in wildtype 2nd instar larvae exposed to a high sugar diet for 8 hours. **B.** *De novo* discovered binding motif (DREME) and its best GATA TF database match (TOMTOM), Grain. **C.** Venn diagram displaying significant overlap between Grain ChIP-seq targets (ENCODE dataset ENCSR909QHH) and upregulated, sugar-dependent genes (RNA-seq, adj.p.val<0.05) in wildtype 2nd instar larvae exposed to a high sugar diet for 8 hours. **D.** Venn diagram showing that the majority of the sugar-responsive genes containing predicted Grain motifs (RNA-seq) in 2nd instar larvae exposed to a high sugar diet for 8 hours are direct targets of Grain (ChIP-seq: ENCODE dataset ENCSR909QHH, E. val<0.05).

influence survival or pupariation kinetics. However, on HSD the pupal development was delayed and survival was significantly compromised (**Figs 2B and S2A and S2 Table**). More-over, the pupal volume of Grain-depleted animals was significantly reduced on both LSD and HSD, being more pronounced on HSD (**Fig 2C**). The delay in development and reduced pupal volume phenotypes on HSD were also observed with an independent RNAi line, albeit with more modest effects (**S2B and S2C Fig and S2 Table**). Both RNAi lines downregulated *grain* expression (**S2C Fig**) and caused full or partial pupal lethality on both diets, respectively (**S2D and S2E Fig**).

## Grain regulates sugar-responsive genes encoding metabolic enzymes and transcription factors

To analyze the role of Grain in sugar-regulated gene expression, we performed an RNA-seq experiment following ubiquitous knockdown of *grain* in whole larvae. For acute high sugar

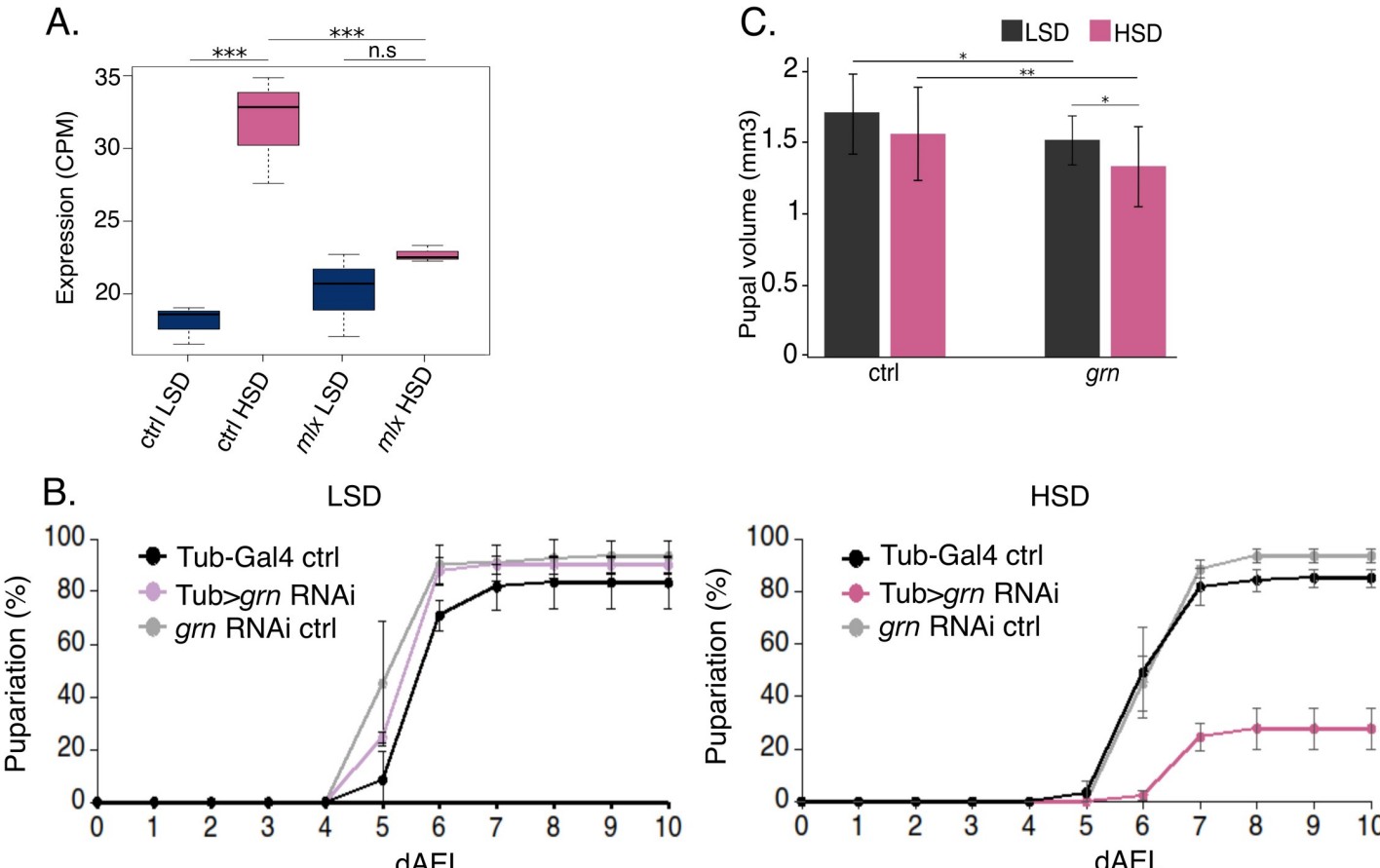

**Fig 2. Loss of Grain leads to sugar intolerance and impaired growth. A.** Sugar-responsive gene expression (RNA-seq) of Grain is inhibited in $mlx^1$ mutant 2nd instar larvae exposed to a high sugar diet for 8 hours (N = 3). **B.** Knockdown of *grain* (Tub-GAL4>*grain* RNAi BDSC #27658) leads to reduced survival and delayed pupariation compared to control genotypes (Tub-GAL4>TriP control BDSC #31603 and UAS-*grain* RNAi BDSC #27658) when raised on high sugar diet (N = 4, 30 larvae per group). **C.** Knockdown of *grain* (Tub-GAL4>*grain* RNAi BDSC #27658) leads to a reduction in pupal volume compared to control larvae (Tub-GAL4>TriP control BDSC #31603) that is aggravated on high sugar diet. (N≥30). Data information: N indicates the number of biological replicates. Error bars display standard deviation. CPM: counts per million. **(A):** *adj.pval<0.05, **adj.p.val<0.01, ***adj.p.val<0.001, CPM: counts per million. **(B).** Log-rank test (Df = 8), p.val.<0.01. **(C):** Two-way ANOVA found no significant interaction effect between diet and genotype. Diet (F (1,146) = 10.176, p.val = 0.00174) and genotype (F (1,146) = 20.430, p. val = 1.27e-05) effects were found to be significantly different. A Tukey HSD was performed with p-values indicated on the graph as follows: *p.val<0.05, **p.val<0.01, ***p.val<0.001. LSD: 10% yeast, HSD: 10% yeast + 15% sucrose.

treatment, the larvae were grown on LSD for 24 h, which was followed by 8 h feeding on either LSD or HSD regimes. To assess the role of Grain in controlling the sugar-activated gene set, we focused on genes that were upregulated in HSD fed larvae, but displayed lower expression in Grain-depleted animals, on either dietary condition. Gene set enrichment of the Grain-regulated sugar-responsive genes revealed significant enrichment of gene sets involved in carbohydrate metabolism and lipid biosynthesis (Fig 3A). Glycolysis was among the most highly enriched gene sets in Grain-deficient animals. Indeed, many glycolytic genes, such as *hexokinase A* (*Hex-A*), *glyceraldehyde-3-phosphate dehydrogenase 1* and *2* (*Gapdh1* & *2*), as well as *pyruvate kinase* (*PyK*) were expressed at a lower level in Grain-deficient animals (Fig 3B). However, the downregulation was mainly observed on LSD, except for *glucose-6-phosphate isomerase* (*Pgi*), which was downregulated by Grain knockdown on both dietary conditions.

To further analyze the possible metabolic role of Grain, we measured the levels of polar metabolites from whole larvae upon Grain depletion on LSD and HSD by using LC-MS-based metabolomics (S1 Data). Several metabolic changes upon Grain depletion were detected

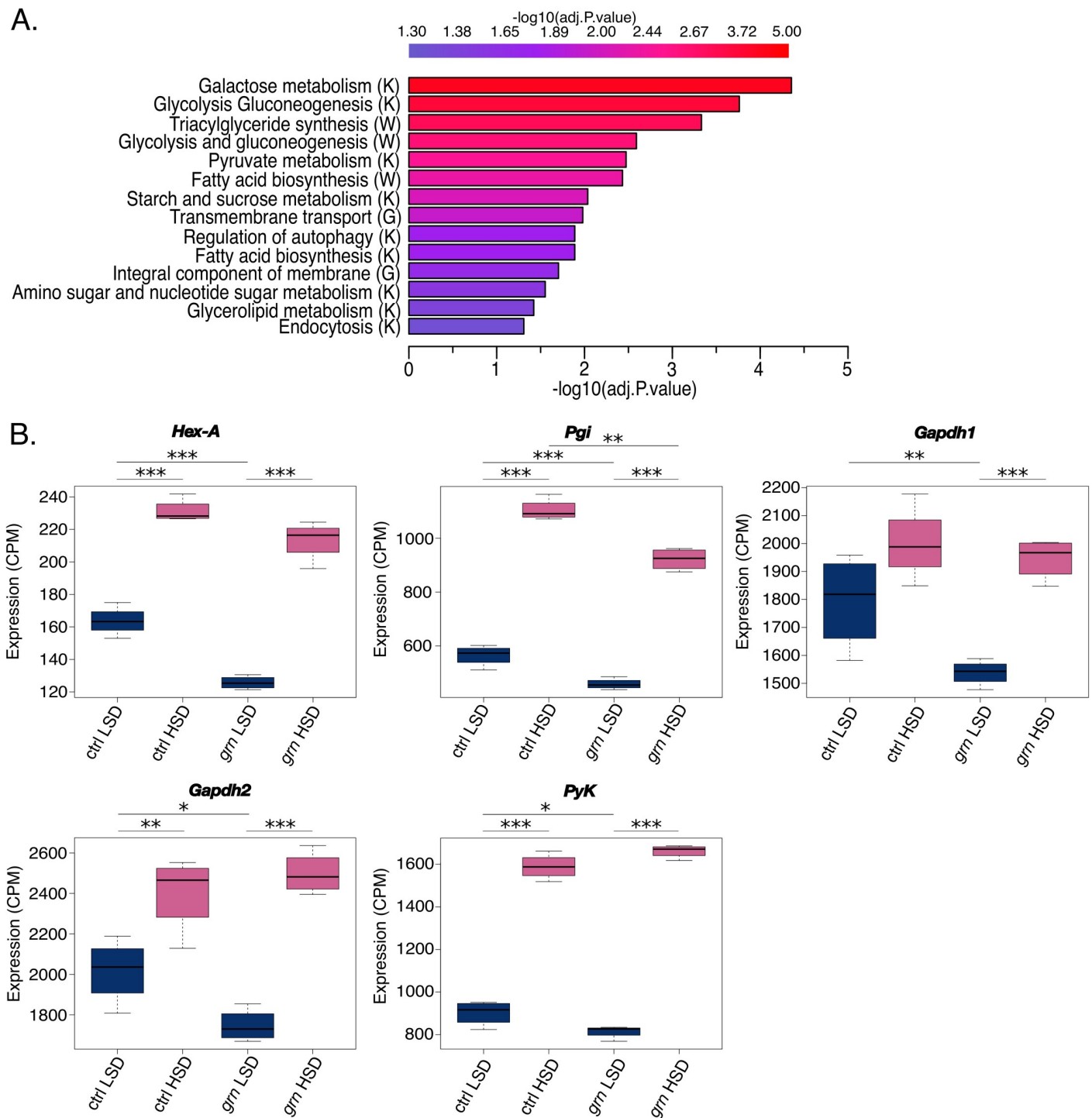

**Fig 3. Grain regulates metabolic gene expression. A.** Gene set enrichment of significantly upregulated, sugar-responsive Grain target genes (RNA-seq, adj.p.val <0.05) in 2nd instar larvae exposed to high sugar diet for 8 hours. Gene sets involved in carbohydrate and lipid metabolism are highly enriched. (hypergeometric test, adj.p.val<0.05). K = KEGG, W = Wikipathways, G = Gene Ontology database. **B.** Expression of several glycolytic genes is downregulated upon *grain* knockdown (Tub-GAL4>*grain* RNAi BDSC #27568) as compared to control (Tub-GAL4>TriP control BDSC #31603) 2nd instar larvae. Data information: *adj.p.val< 0.05, **adj.p.val<0.01, ***adj.p.val<0.001. CPM: counts per million. LSD: 10% yeast, HSD: 10% yeast + 15% sucrose.

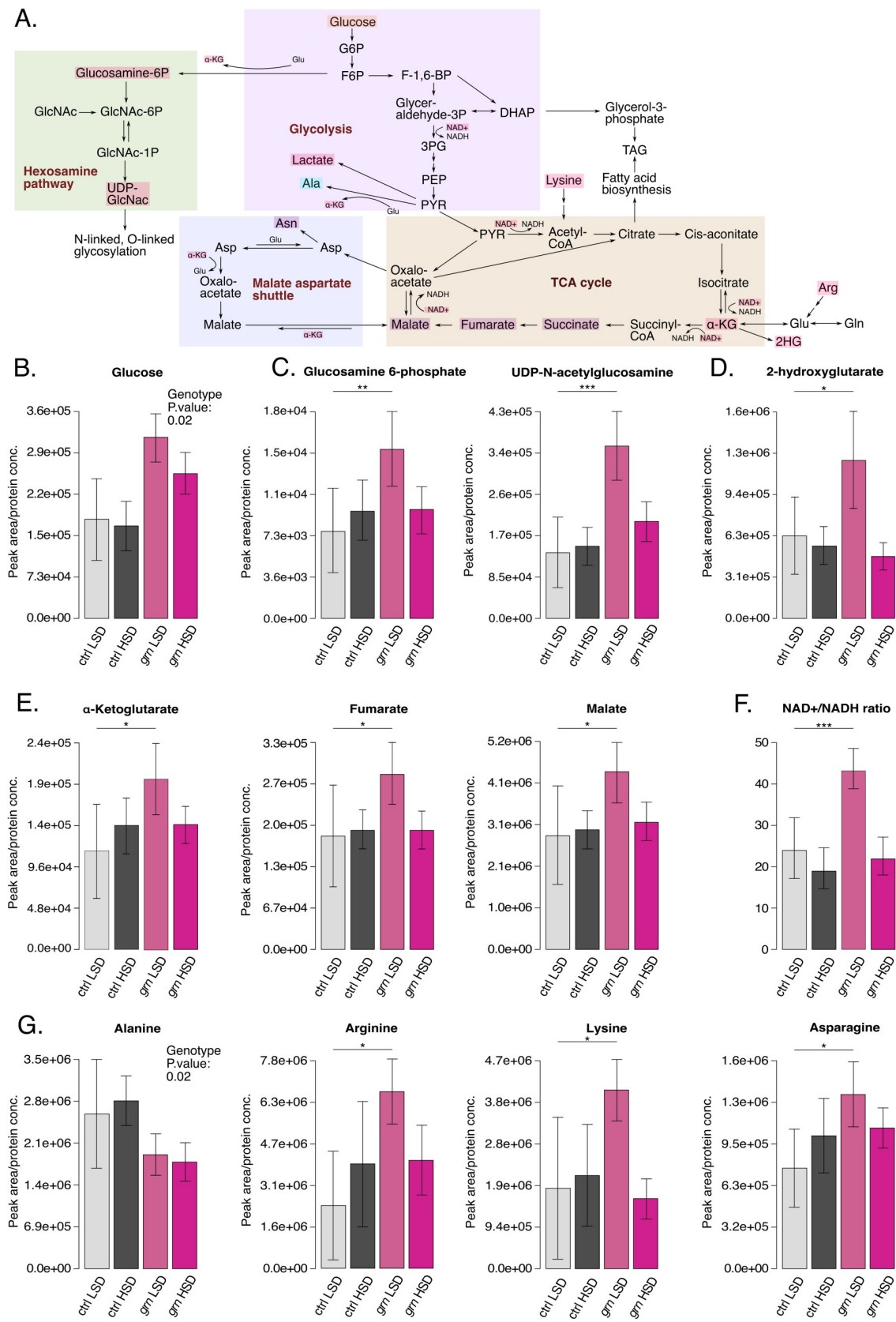

**Fig 4. Grain controls central carbon metabolism. A.** Global overview of selected metabolic pathways (LC-MS analysis of polar metabolites). Metabolites that were significantly elevated (Two-way ANOVA, genotype:diet interaction, $F_{(1,15)} > 5.0$, p.val < 0.05) upon *grain* knockdown (Tub-GAL4>*grain* RNAi BDSC #27658) compared to control (Tub-GAL4>TriP control BDSC #31603) 2nd instar larvae exposed to a high sugar diet for 24 hours are highlighted in red, borderline significantly elevated (two-way ANOVA, genotype:diet interaction, $F_{(1,15)} > 3.5$, p.val < 0.08) in purple. While some metabolites didn't reach significant interaction, their genotype effect (two-way ANOVA, genotype, $F_{(1,15)} > 7.1$, p.val < 0.05) was significant. These metabolites are marked in orange (upregulated) and blue (downregulated). Full statistical data found in S1 Data. **B—E.** Grain depletion (Tub-GAL4>*grain* RNAi BDSC #27658) leads to elevated levels of several metabolites, including Glucose (two-way ANOVA, genotype, $F_{(1,15)} = 7.13$, p.val = 0.017) (**B**), hexosamine pathway intermediates (two-way ANOVA, genotype:diet, $F_{(1,15)} > 6.32$, p.val < 0.05) (**C**), 2-hydroxyglutarate (two-way ANOVA, genotype:diet, $F_{(1,15)} = 7.50$, p.val = 0.015) (**D**), and intermediates of the TCA cycle with borderline significance (two-way ANOVA, genotype:diet, $F_{(1,15)} > 3.54$, p.val < 0.08) (**E**) on low sugar diet compared to control (Tub-GAL4>TriP control BDSC #31603). **F.** The $NAD^+$/NADH balance is elevated upon Grain depletion (Tub-GAL4>*grain* RNAi BDSC #27658) on low sugar diet as compared to control (Tub-GAL4>TriP control BDSC #31603) (two-way ANOVA, genotype:diet, $F_{(1,15)} = 9.90$, p.val = 0.007). **G.** Changes in amino acid levels upon Grain depletion (Tub-GAL4>*grain* RNAi BDSC #27658) compared to control (Tub-GAL4>TriP control BDSC #31603). 2nd instar larvae exposed to a high sugar diet for 24 hours (LC-MS analysis of polar metabolites). (Arg, Lys, Asn: two-way ANOVA, genotype:diet, $F_{(1,15)} > 4.50$, p.val < 0.06). Data information: N indicates the number of biological replicates. B-G) For all metabolites (apart from glucose and alanine) a two-way ANOVA revealed at least a borderline significant interaction effect between diet and genotype ($F_{(1,15)} > 3.5$, p.val < 0.08). A Tukey HSD post hoc analysis revealed this interaction to be caused by an increase of these metabolites upon *grn* knockdown on a low sugar diet (p values indicated on graphs: *p.val< 0.05, **p.val<0.01, ***p.val<0.001, df grn HSD-ctrl LSD = 8). Glucose and alanine did not reach a significant interaction effect, but showed significant genotype effects ($F_{(1,15)} > 7.1$, p.val < 0.05). Full statistical data found in Data S1. * LSD: 10% yeast, HSD: 10% yeast + 15% sucrose. N = 5 for control LSD and HSD and *grain* RNAi LSD, 4 for *grain* RNAi HSD, 20 larvae per replicate.

(**Figs 4A and S3**). Grain depletion increased the levels of glucose on both dietary conditions (**Fig 4B**). Levels of metabolites that branch out of the glycolytic pathway and TCA cycle were elevated on LSD. These include intermediates of the hexosamine pathway (Glucosamine 6-phosphate, UDP-N-acetylglucosamine) (**Fig 4C**), as well as 2-hydroxyglutarate (**Fig 4D**), which is synthetized by *lactate dehydrogenase* in *Drosophila* [26]. Several intermediates of the TCA cycle displayed moderately elevated levels on LSD, with borderline statistical significance (**Fig 4E**). Grain-dependent metabolic changes were also reflected to redox balance, as $NAD^+$/NADH balance was highly elevated upon Grain depletion on LSD (**Fig 4F**). Amino acids arginine, lysine and asparagine were modestly elevated by Grain depletion on LSD, while alanine levels were modestly downregulated by *grain* knockdown (**Fig 4G**). To conclude, Grain regulates central carbon and amino acid metabolism, in particular on LSD.

In addition to metabolic genes, we identified sugar-responsive TF genes among Grain targets. Cabut is a transcriptional repressor, which is directly activated by Mondo-Mlx upon sugar feeding [5,12]. Interestingly, *cabut* expression was significantly downregulated upon Grain depletion in LSD fed animals (**Fig 5A**). Moreover, the ENCODE-derived embryonic ChIP-seq data revealed that *cabut* is a direct target of Grain (**Fig 5B**). The expression of Activin/Dawdle downstream TF *smox* was also found to be activated by sugar, but its expression was significantly downregulated by Grain depletion upon LSD feeding (**Fig 5C**). Similar to *cabut*, *smox* was identified as a direct Grain target in the ENCODE ChIP-seq data (**Fig 5D**). In addition to the TFs involved in sugar tolerance, Grain-depleted animals displayed significant downregulation of Aldehyde dehydrogenase *Aldh-III*, which is also a direct Grain target (**S4 Fig**). Notably, we have previously shown that Aldh-III is necessary for sugar tolerance [5]. In conclusion, Grain regulates the expression of metabolic genes as well as TFs with a role in metabolic regulation and sugar tolerance.

## Grain converges with Sugarbabe to regulate lipid metabolism

The discovery of *cabut* and *smox* as targets of Grain led us to systematically analyze the possible overlap of other Grain targets and those of Sugarbabe and Mlx, the other known sugar-responsive TFs. Indeed, sugar-responsive target genes of both Sugarbabe and Mlx showed

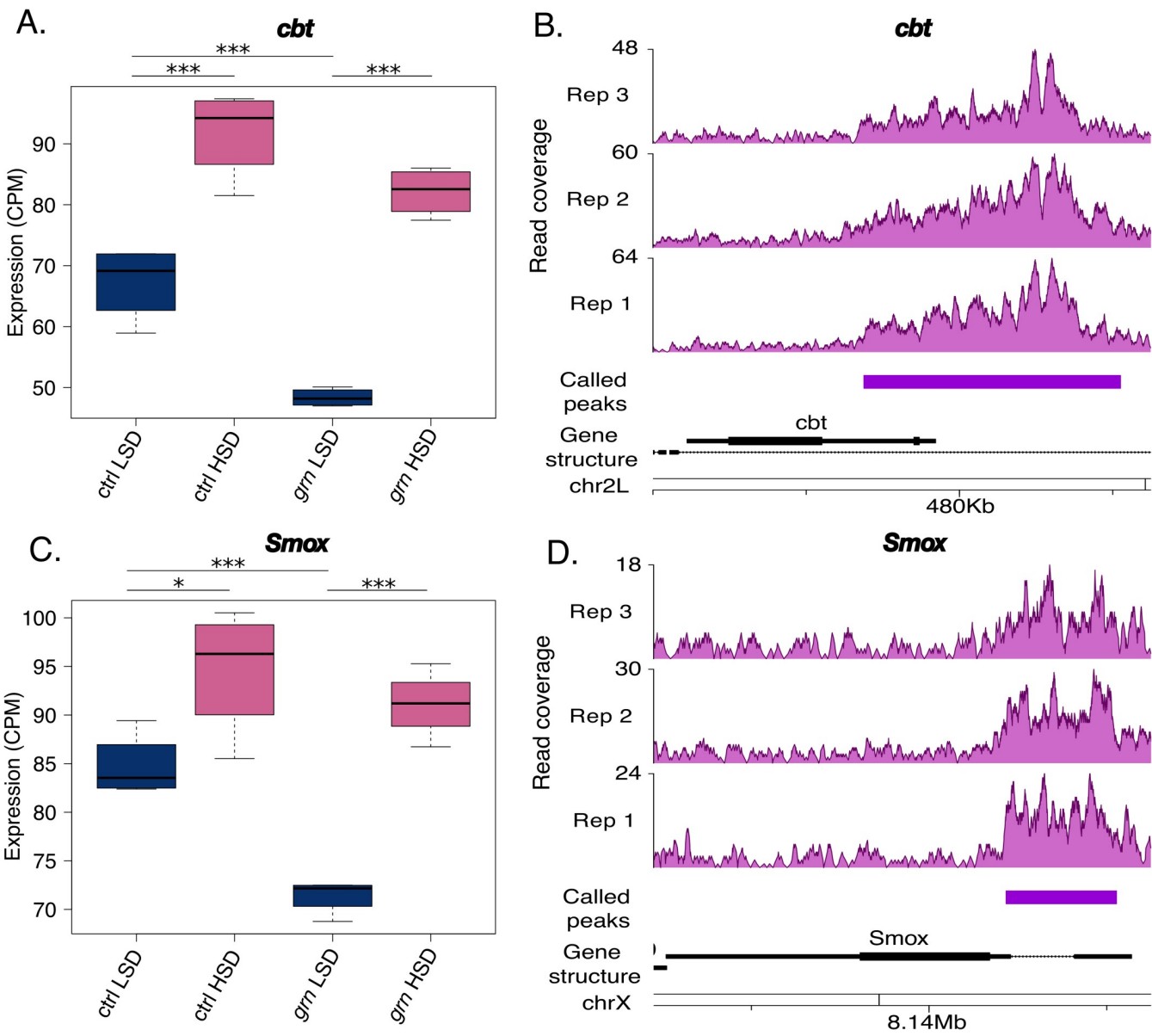

**Fig 5. Sugar-responsive transcription factors Cabut and Smox are regulated by Grain. A.** mRNA expression (RNA-seq) of transcription factor *cabut* (*cbt*) is downregulated upon *grain* knockdown (Tub-GAL4>*grain* RNAi BDSC #27658) as compared to control (Tub-GAL4>TriP control BDSC #31603) 2^nd instar larvae on low sugar diet (N = 4, 20 per group). **B.** Binding profile of Grain in *cabut* promoter (ChIP-seq, ENCODE dataset ENCSR909QHH). The purple bar indicates the location of the called *cabut* peak in the dataset. **C.** mRNA expression (RNA-seq) transcription factor *smox* is downregulated upon *grain* knockdown (Tub-GAL4>*grain* RNAi BDSC #27658) as compared to control (Tub-GAL4>TriP control BDSC #31603) 2^nd instar larvae on a low sugar diet (N = 4, 20 per group). **D.** Binding profile of Grain in *smox* promoter (ChIP-seq, ENCODE dataset ENCSR909QHH). The purple bar indicates the location of the called *smox* peak in the dataset. Data information: N indicates the number of biological replicates. *adj.p.val<0.05, ***adj.p.val<0.001. LSD: 10% yeast, HSD: 10% yeast + 15% sucrose. CPM: counts per million.

significant overlap with those of Grain (**Figs 6A and S5A**). 66 genes that were activated by sugar feeding were downregulated both in Sugarbabe-deficient animals and upon Grain depletion (**Fig 6A**). Collective analysis of the expression of this set of common target genes (displayed as heatmaps) showed that Grain depletion most prominently downregulated the gene expression in LSD feeding animals, while a subset of genes was also downregulated on HSD (**Fig 6B**). On the other hand, Sugarbabe mainly contributed to the regulation of these target

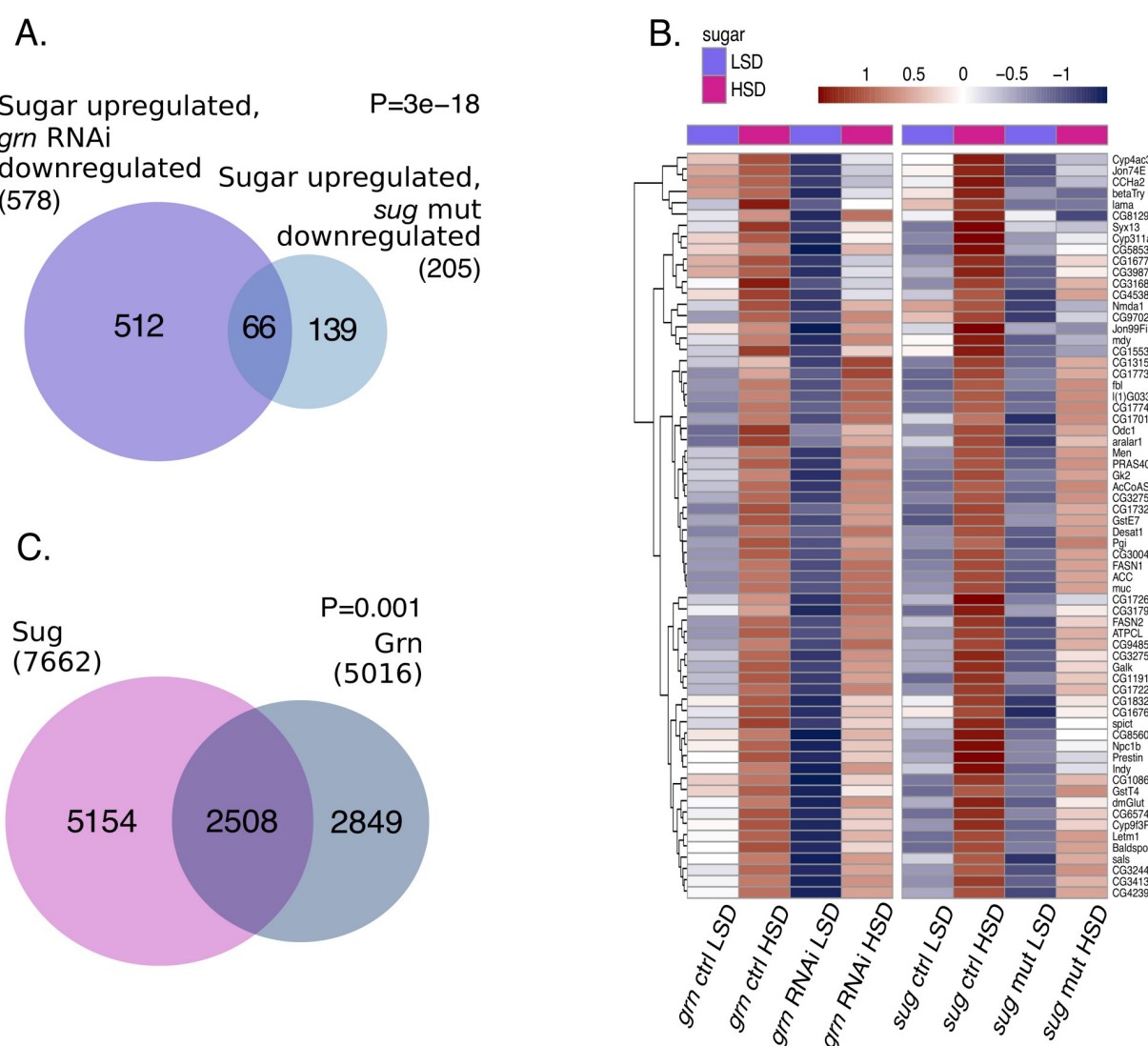

**Fig 6. Grain and Sugarbabe share common target genes. A.** Venn diagram displaying overlap between sugar-activated, Grain-dependent targets (adj.p.val<0.05) and sugar-activated, Sugarbabe-dependent targets (adj.p.val<0.05) determined by RNA-seq in 2nd instar larvae exposed to HSD for 8 hours. **B.** Heatmaps displaying the overlapping genes in (**A**). Shared target genes of Grain (Grn) and Sugarbabe (Sug) are primarily regulated by Sugarbabe on high sugar diet, and regulated by Grain on low sugar diet. Color key displays scaled log2 gene expression. LSD: 10% yeast, HSD: 10% yeast + 15% sucrose. **C.** Venn diagram displaying overlap (>50% overlap in called peak areas) between direct targets of Grain and Sugarbabe (ChIP-seq, ENCODE datasets ENCSR909QHH and ENCSR008NYP).

genes upon HSD feeding. Grain knockdown did not influence *sugarbabe* expression (**S5B Fig**).

As Grain shared several target genes with Sugarbabe, we systematically compared the ENCODE-derived ChIP-seq data on direct Grain and Sugarbabe targets, looking for peaks that displayed >50% overlap in the *Drosophila* genome. Interestingly, we found over 2500 overlapping Grain and Sugarbabe genomic binding sites (**Fig 6C**). Common targets of Grain and Sugarbabe included genes that function in the interface of central carbon metabolism and fatty acid biosynthesis, such as *Acetyl-CoA carboxylase* (*ACC*), and *Fatty acid synthase* (*FASN1*), *ATP citrate lyase* (*ATPCL*), *Acetyl-CoA synthetase* (*AcCoAS*) (**Fig 7A**). These lipogenic genes displayed highly similar binding profiles for Grain and Sugarbabe (**Fig 7A**). Their

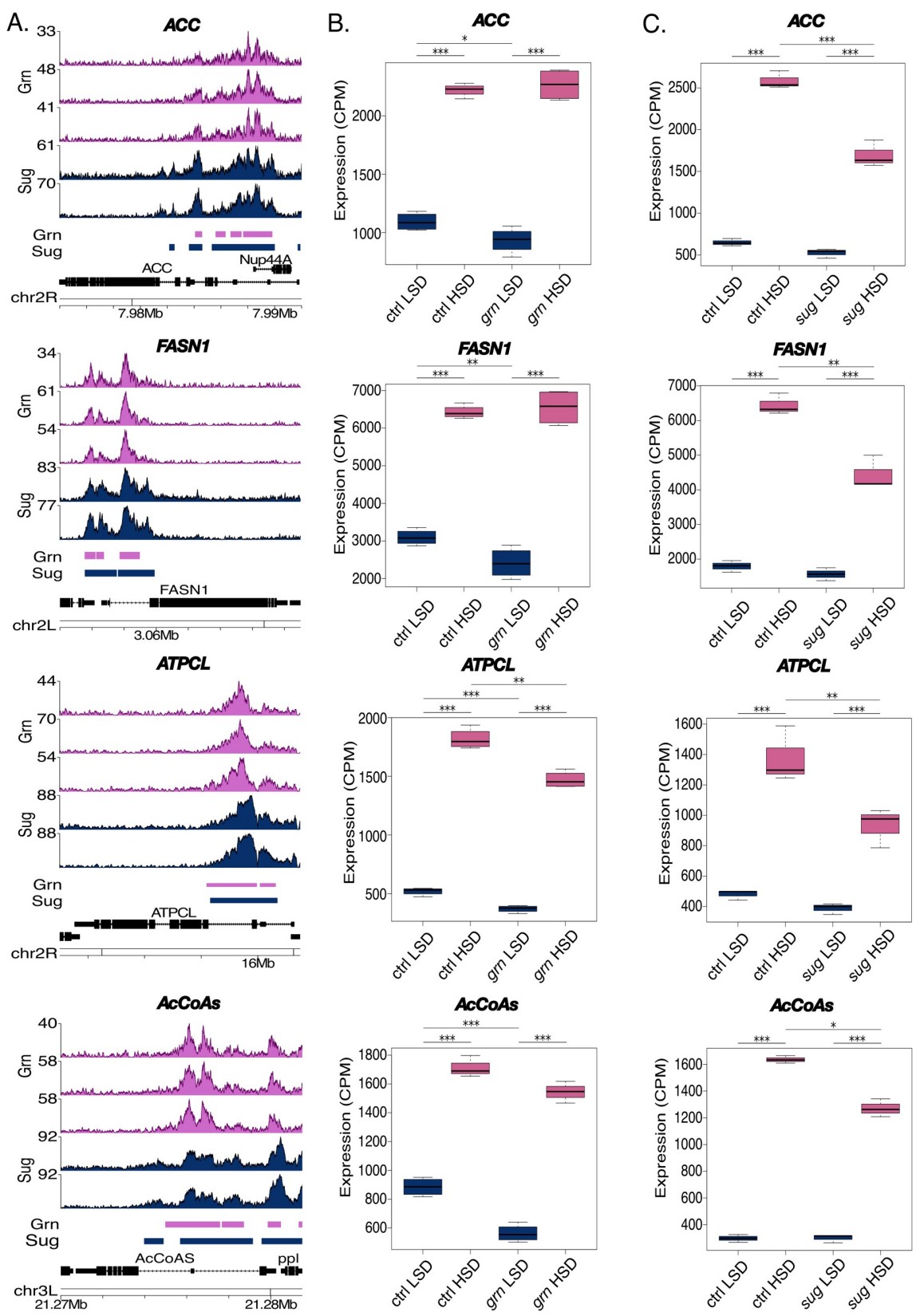

**Fig 7. Grain and Sugarbabe co-regulate genes involved in lipid metabolism. A.** Binding profiles of Grain and Sugarbabe (ChIP-seq, ENCODE datasets ENCSR909QHH and ENCSR008NYP) display overlapping binding sites in the promoters of lipogenic genes. Pink peaks and bar indicate the expression and called peak location in Grain, whereas dark blue denotes the same for Sugarbabe. **B and C.** Grain and Sugarbabe regulate the expression of genes (RNA-seq) involved in *de novo* lipogenesis. mRNA expression of *ACC*, *FASN1*, *ATPCL* and *AcCoAs* in *grain* knockdown (Tub-GAL4>*grain* RNAi BDSC #27658) and control (Tub-GAL4>TriP control BDSC #31603) 2$^{nd}$ instar larvae on LSD or exposed to HSD for 8 hours. (N = 4, 20 per group) (**B**), compared to expression of the same genes in *sug* mutants (6) (**C**). Data information: N indicates the number of biological replicates. *adj.p.val<0.05, **adj.p.val<0.01, ***adj.p.val<0.001. LSD: 10% yeast, HSD: 10% yeast + 15% sucrose. CPM: counts per million.

expression was significantly downregulated in animals deficient for these TFs, Grain contributing mainly on LSD and Sugarbabe on HSD (**Fig 7B and 7C**) [6].

Consistent with the reduced lipogenic gene expression, we observed a moderate reduction of triacylglycerol (TAG) levels in Grain-depleted larvae grown on LSD (**Fig 8A**). This was confirmed with lipidomics analysis of whole larvae, showing reduced TAG/PE (phosphatidylethanolamine) ratio upon Grain depletion on LSD (**Fig 8B**). Interestingly the lipidomics analysis also revealed a significant increase in TAG acyl chain length upon Grain depletion, which is in line with reduced larval total lipogenesis producing short *de novo* synthesized acyl chains (**Figs 8C and S6A**) [27]. Analysis of lipid droplets in larval fat body showed significantly reduced levels of normalized lipid droplet volumes in Grain-depleted animals on LSD (**Figs 8D and 8E and S6B**), consistent with the biochemical TAG measurements and the observed changes in lipogenic gene expression.

## Tissue-specific analysis reveals sugar-dependent and independent functions of Grain

Our data shows that while *grain* displays sugar-inducible expression and contributes to growth and survival on HSD, it controls central carbon and lipid metabolism as well as metabolic gene expression primarily on LSD. To better understand the underlying reasons, we analyzed the tissue-specific roles of *grain*. The data from *Drosophila* gene expression database FlyAtlas2 showed that *grain* mRNA expression was highest in the midgut (**Fig 9A**). High midgut expression was confirmed by quantitative RT-PCR analysis of RNA isolated from four larval tissues (**S7A Fig**). Using FlyAtlas2 data we also calculated the relative tissue-specific expressions of all genes in the gene set of **Fig 3A**. Consistently with high *grain* expression in the midgut, the mean of these relative expression levels shows that genes upregulated by sugar and downregulated by *grain* knockdown displayed highest expression in the midgut (39.5%), followed by the fat body and Malpighian tubules (**S7B Fig**). This was in contrast to all genes present in FlyAtlas2 with 11.9% mean relative expression in the midgut (**S7C Fig**).

Next we analyzed the contribution of sugar feeding on *grain* and its targets in tissue-specific manner. Interestingly, sugar-induced Grain expression was only observed in the fat body and Malpighian tubules, but not in the midgut or carcass (**Fig 9B**). Consistent with this finding, ubiquitous Grain knockdown blunted sugar-induced expression of several target genes in the fat body, including TF genes *cabut* and *smox* (**Fig 9C**), as well as lipogenic genes, *ATPCL*, *AcCoAS*, *ACC*, and *FASN1* (**Fig 9D**). In contrast to fat body, Grain did not inhibit sugar-induced lipogenic gene expression in the midgut, but displayed reduced expression under LSD conditions (**Fig 9D**). This was interesting, considering the observed TAG phenotypes on LSD and the critical role of midgut in larval lipid biosynthesis [28].

Finally, we analyzed the function of Grain in the larval midgut more closely. The use of a *grain* GFP fusion (PBac{grn-GFP.FPTB}) revealed that *grain* displays highly specific expression pattern, showing prominent expression in the anterior-most parts of the midgut, along with the gastric caeca (**Fig 10A**). Interestingly, the anterior-most area of the midgut has earlier

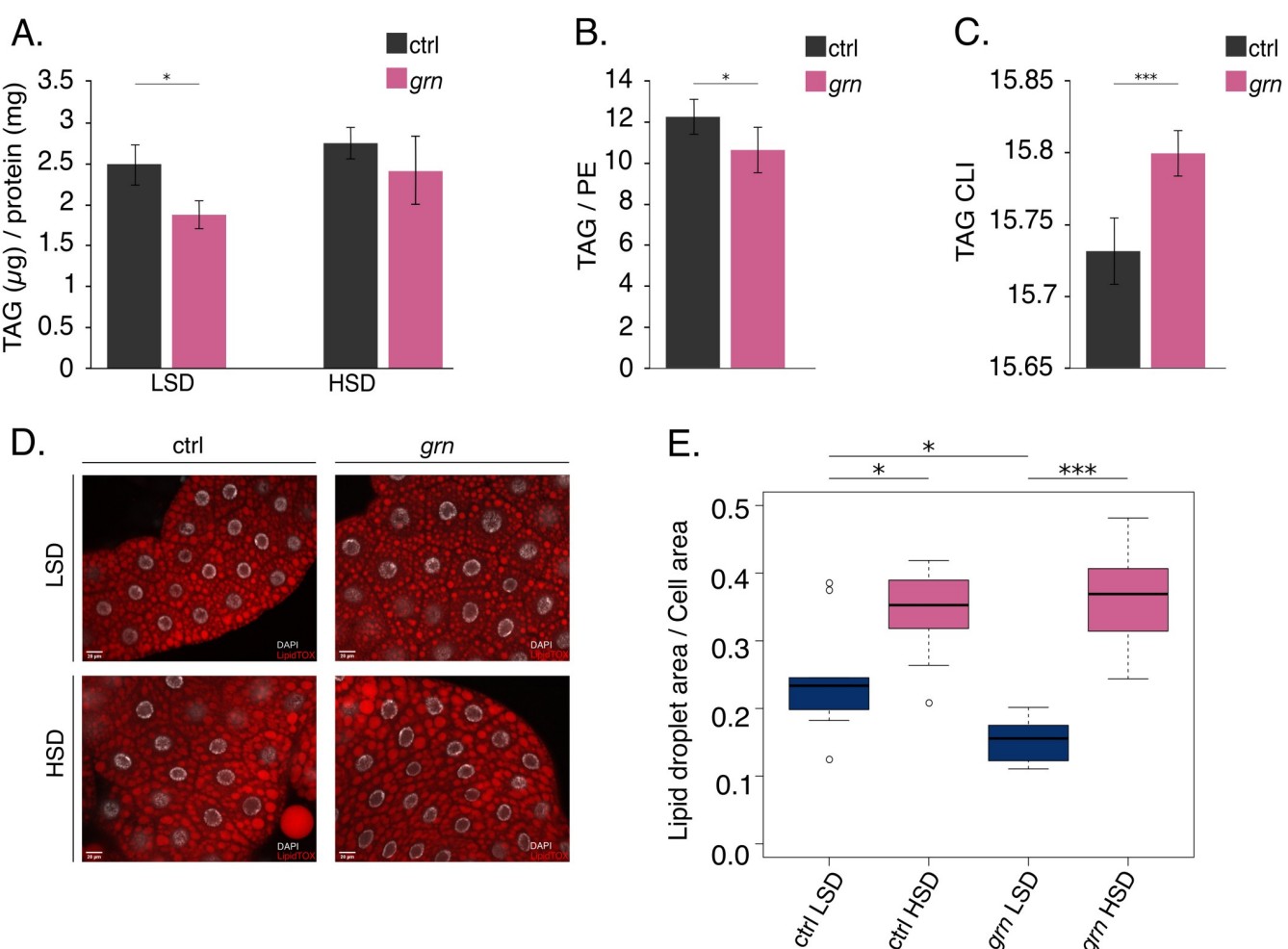

**Fig 8. Grain depletion leads to disturbed lipid biosynthesis. A.** Grain depletion (Tub-GAL4>*grain* RNAi BDSC #27658) leads to a reduction in TAG stores on a diet low in sugar compared to control early 3rd instar larvae (Tub-GAL4>TriP control BDSC #31603). Enzymatic TAG assay, normalized to protein levels (N = 4, 10 larvae per group). **B.** Grain depletion (Tub-GAL4>*grain* RNAi BDSC #27658) leads to a reduction in total TAG/PE molar ratios (measured by mass spectrometry-based lipidomics,) compared to control (Tub-GAL4>TriP control BDSC #31603) pre-wandering 3rd instar larvae after being fed glucose for 24 hours. (N = 4, 15 larvae per sample). **C.** Grain-depleted larvae had in their TAG on average longer acyl chains (expressed as CLI representing the average length of one acyl chain) than the control larvae (N = 4, 15 larvae per sample). **D.** Representative immunofluorescence images of lipid droplet (LipidTOX) and nuclei (DAPI) staining in Grain depleted (Tub-GAL4>*grain* RNAi BDSC #27658) and control (Tub-GAL4>TriP control BDSC #31603) fat bodies of early 3rd instar larvae raised on a diet either low or high in sugar. **E.** Quantification of lipid droplet staining in **D**. Fat bodies from Grain depleted (Tub-GAL4>*grain* RNAi BDSC #27658) 3rd instar larvae have lower relative amount of lipid droplets (total lipid droplet area/cell area) than controls (Tub-GAL4>TriP control BDSC #31603) (N = 9 cells for each sample group). Data information: N indicates the number of biological replicates. Error bars display standard deviation. **(A):** Two-way ANOVA found no significant interaction effect between diet and genotype. Diet (F (1,12) = 8.381, p.val = 0.01345) and genotype (F (1,12) = 11.602, p.val = 0.00521) effects were found to be significantly different. A Tukey HSD was performed with p-values indicated on the graph as described below. **(B):** one-tailed Student's t-test (6 degrees of freedom), **(C):** two-tailed Student's t-test (6 degrees of freedom). **(E)** Two-way ANOVA found a significant interaction effect between diet and genotype (F (1,32) = 5.262, p.val = 0.0285). A Tukey HSD was performed with p-values indicated on the graph as follows: *p.val< 0.05, **p.val<0.01,***p.val<0.001. LSD: 10% yeast, HSD: 10% yeast + 15% sucrose, TAG: triacylglycerol, PE: phosphatidylethanolamine, CLI: chain length index.

been shown to possess high lipogenic activity [28]. To test the functional importance of Grain in the intestine, we analyzed the expression of *ACC* by using a LacZ reporter upon Grain knockdown. Intriguingly, the expression of the reporter-encoded β-Galactosidase in the anterior midgut was prominently inhibited upon Grain knockdown on both LSD and HSD conditions (**Fig 10B and 10C**). In conclusion, Grain displays tissue-specific expression and

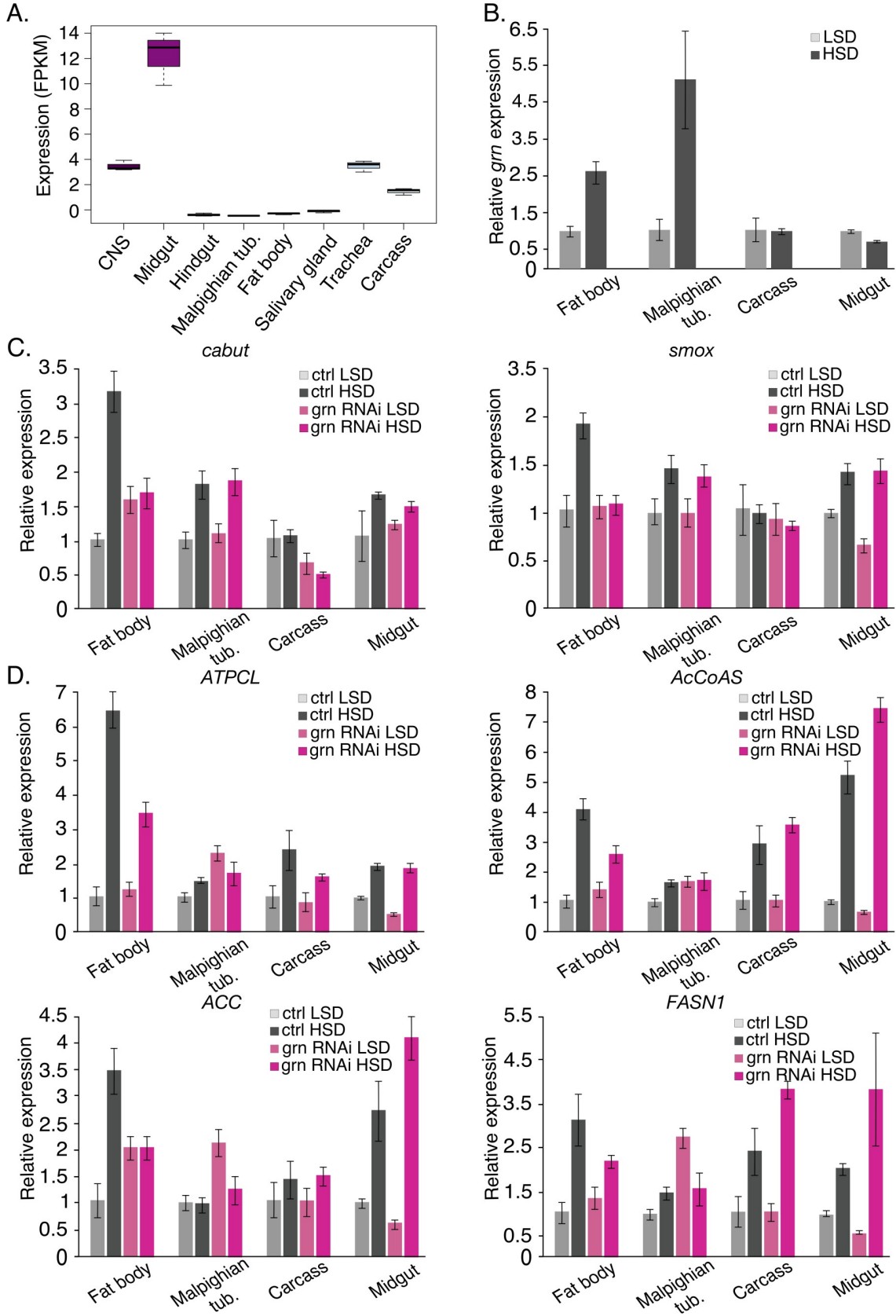

**Fig 9. Tissue specific expression and function of *grain*. A.** mRNA expression (RNAseq, FPKM: fragment per kilobase million) of *grain* (*grn*) in larval tissues according to FlyAtlas2. **B.** Normalized tissue expression of *grain* (*grn*) mRNA in larval fat body, Malpighian tubules, carcass, and midgut of *grain* knockdown (Tub-GAL4>*grain* RNAi BDSC #27658) and control (Tub-GAL4>TriP control BDSC #31603) early 3rd instar larvae on LSD or exposed to HSD for 8 hours. Expression of *CDK7* was used for normalization. **C.** Normalized tissue expression of Cabut and Smox mRNA of *grain* knockdown (Tub-GAL4>*grain* RNAi BDSC #27658) and control (Tub-GAL4>TriP control BDSC #31603) early 3rd instar larvae on LSD or exposed to HSD for 8 hours. Expression of *CDK7* was used for normalization. **D.** Normalized tissue expression of lipogenic gene mRNA of *grain* knockdown (Tub-GAL4>*grain* RNAi BDSC #27658) and control (Tub-GAL4>TriP control BDSC #31603) early 3rd instar larvae on LSD or exposed to HSD for 8 hours. Expression of *CDK7* was used for normalization. Data information: Data in panels B-D represents cDNA pools of 4 biological replicates (tissues from 5 animals/replicate). Error bars display standard deviation of technical replicates. LSD: 10% yeast, HSD: 10% yeast + 15% sucrose.

function, contributing to sugar-inducible gene expression in the fat body while displaying high constitutive expression in the midgut to promote lipogenic gene expression.

## Discussion

We provide evidence for a role for GATA transcription factor Grain in metabolic gene regulation *in vivo* (**Fig 11**). This discovery was initiated by an unbiased transcription factor (TF) target motif enrichment analysis, which revealed a significant over-representation of GATA-like motifs in the promoter regions of sugar-regulated genes in *Drosophila* larvae. Subsequent experimental and computational validation singled out Grain as the TF involved. Specifically, we discovered that: 1) Grain regulates expression of metabolic genes, including glycolytic and lipogenic genes, 2) Grain also controls the expression of genes encoding other sugar-responsive TFs, such as Cabut and Smox, 3) Grain controls sugar-responsive target gene expression in the fat body, while having a constitutive regulatory role in the anterior midgut, 4) Grain shares genomic binding sites and target genes with sugar-activated transcription factor Sugarbabe, 5) Grain and Sugarbabe predominate on the regulation of common targets on low and high sugar, respectively, 6) Grain-deficient larvae display disturbed growth on HSD, 7) Grain deficient animals display elevated levels of glucose-derived metabolic intermediates, such as those of the hexosamine pathway, but lower TAG stores on LSD. Collectively, our data provides evidence for Grain, the *Drosophila* ortholog of GATA1/2/3 subfamily, as a tissue-specific regulator of nutrient-responsive gene expression.

Earlier studies on Grain have established its roles in developmental processes of *Drosophila*, including the regulation of organ morphogenesis [18] as well as axon guidance through the regulation of cell adhesion molecules [20,29]. In mammals, the orthologous GATA-2 and GATA-3 TFs indirectly influence organismal energy metabolism through the control of adipocyte terminal differentiation [21]. Our findings show that in addition to affecting metabolism through differentiation, Grain has a direct role in controlling the expression of genes encoding metabolic enzymes. More specifically, Grain integrates into the gene regulatory network that responds to changes in dietary sugars and whose master regulator is the sugar sensing Mondo-Mlx heterodimer. Considering the differential requirements for developmental and nutrient-responsive gene regulation, such dual role may seem surprising. Notably, however, earlier studies have uncovered other TFs with similarly diverse physiological roles. For example, the *Drosophila* Klf-10 ortholog Cabut, a direct target of Grain, serves as a Mondo-Mlx-dependent transcriptional repressor in the sugar responsive gene regulatory network [5,12] along with developmental roles in dorsal closure and wing patterning [30,31]. How such multifunctional TFs achieve the necessary target gene specificity remains unresolved. One possible mechanism is through cooperative binding with other transcription factors with more specific physiological role.

Comparative analysis of ENCODE ChIP-seq data revealed extensive overlap between genomic binding sites of Grain and Sugarbabe, raising the possibility for mechanistic interaction

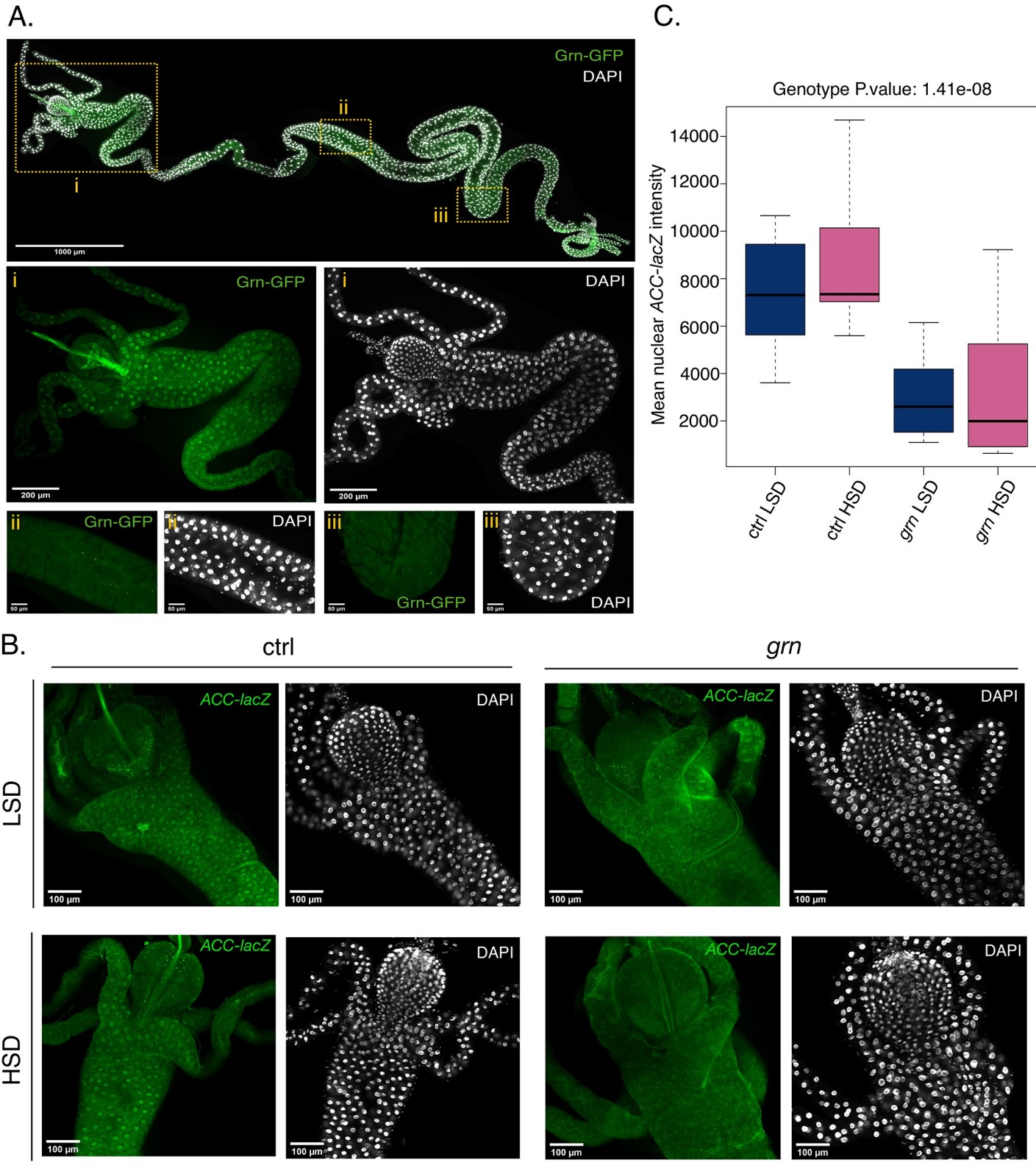

**Fig 10. Grain regulates lipogenic gene expression in the anterior midgut. A.** Grain-GFP is highly expressed in the anterior parts of the larval midgut. Representative images of midguts with nuclear staining (DAPI) of early 3$^{rd}$ instar larvae expressing Grain-GFP (BDSC #58483) on LSD. **B.** Grain knockdown inhibits the expression of *ACC lacZ* reporter in the anterior midgut. Representative immunofluorescence images of midguts of early 3$^{rd}$ instar *grain* knockdown (Tub-GAL4>*grain* RNAi BDSC #27658) and control (Tub-GAL4>TriP control BDSC #31603) larvae expressing nuclear β-Galactosidase under the *ACC* promoter (BDSC #10790), exposed to LSD or HSD for 8 hours. ACC LacZ is stained with anti-β-Galactosidase and DNA is stained with DAPI. **C.** Quantification of mean nuclear ACC LacZ intensity in **B.** Cells in the anterior midgut of Grain depleted (Tub-GAL4>*grain* RNAi BDSC #27658) 3$^{rd}$ instar larvae express lower levels of ACC LacZ than controls (Tub-GAL4>TriP control BDSC #31603) (N = 12 nuclei for each sample group). Data information: (**C**): Two-way ANOVA found no significant interaction effect between diet and genotype. A genotype (F (1,44) = 48.123, p.val = 1.41e-08) effect was found to be significantly different. LSD: holidic medium without sucrose (64), HSD: holidic medium with added 15% sucrose.

between these TFs. While the resolution of ChIP-seq did not allow precise determination of the topology of the binding sites in this study, it will be interesting to further dissect the possible interplay between the chromatin binding of these two transcription factors in the future. Sugarbabe is a direct target of Mondo-Mlx and a downstream target of Dawdle/Smox-mediated Activin signaling [6] and it is among the most strongly sugar-activated genes in *Drosophila* larvae [13]. Its known physiological roles are related to metabolic control, therefore having

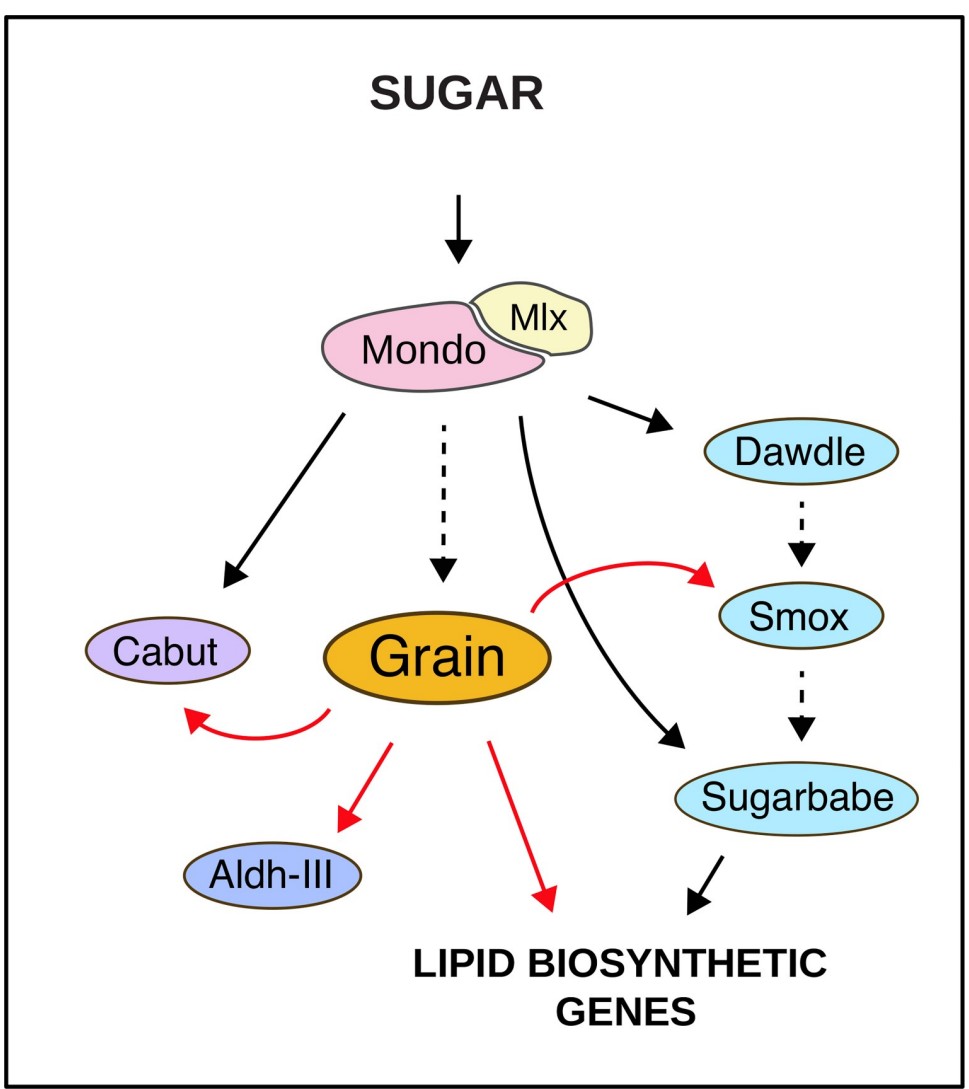

**Fig 11. Model of Grain function in sugar responsive gene regulatory network.**

the characteristics of a specificity determining factor for Grain in the context of metabolic gene regulation. Our RNA-seq analyses revealed that a significant number of key metabolic target genes, including those involved in *de novo* lipogenesis, were regulated by both factors. The regulatory impact of Grain was more prominent at the low end of dietary sugar spectrum, while Sugarbabe displayed functional predominance on high sugar feeding. Similar differences were seen in the regulation of TAG storage, with Grain having a role on LSD conditions and Sugarbabe promoting the increase of TAG levels also on HSD [6]. Considering the findings on the effects of dietary sugar *vs.* protein balance on TAG storage of *Drosophila* [32], it will be interesting to further explore how activities of Grain and Sugarbabe modulate the diet response surface in this setting. Alternatively, it is possible that Grain and Sugarbabe act independently to control overlapping target genes in different tissue locations, such as distinct intestinal regions. Grain displayed highest expression levels in the midgut, being particularly strong in the anterior region. Experiments with *ACC* reporter revealed prominent Grain activity in the anterior midgut, where Grain maintained *ACC* expression under both LSD and HSD. This strong, but localized, constitutive Grain activity might explain the gene expression patterns observed at the level of whole animal, with Grain dependent expression mostly on LSD. It is possible that these localized effects of Grain are overridden by strong sugar-inducible activation of Mondo-Mlx and Sugarbabe in other regions of the midgut. This highlights the need for regionally-resolved analysis of sugar responsive gene expression in the midgut.

Grain expression was increased and the Grain-depleted animals displayed a growth inhibition on HSD, which contrast with the findings on the primary metabolic role for Grain on LSD. Our tissue-specific gene expression analysis provided a possible explanation for this apparent discrepancy. We observed that *grain* expression was sugar responsive in specific tissues, such as fat body, while having high constitutive expression in the midgut. Grain-mediated regulation of target genes followed this tissue-specific pattern. Thus, the observed HSD and LSD-specific physiological phenotypes might be consequences of these different tissue-specific roles, respectively. In addition to the deregulation of glycolytic and lipogenic genes, we also observed a prominent downregulation of Aldh-III, which we have earlier established as a strong determinant of sugar tolerance [5]. Collectively, these data support our earlier findings that the regulation of central carbon and lipid metabolism can, at least in specific circumstances, be uncoupled from sugar tolerance. Future studies, employing genetic epistasis experiments, are needed to test the contribution of Grain target genes on the sugar tolerance phenotype. In conclusion, our study establishes a metazoan GATA transcription factor as a regulator of metabolic gene expression and consequently adds a new member to the gene regulatory network that controls energy metabolism and sugar tolerance in response to changing dietary sugar intake.

## Materials and methods

### *Drosophila* stocks and husbandry

All fly stocks were maintained on a standard laboratory diet containing 6.5% malt (w/v), 3.2% semolina (w/v), 1.8% dry baker's yeast (w/v), 0.6% agar (w/v), 0.7% propionic acid (v/v) and 2.5% Nipagin (methyl paraben) (v/v) (5) at 25°C on a 12 hours light– 12 hours dark light cycle. Experiments were performed on simple dietary regimes, low sugar diet (LSD, 10% dry baker's yeast (w/v), 0.5% agarose (w/v), 0.7% propionic acid (v/v) and 2.5% Nipagin (v/v)) and high sugar diet (HSD, 10% dry baker's yeast (w/v), 15% sucrose (w/v), 0.5% agarose (w/v), 0.7% propionic acid (v/v) and 2.5% Nipagin (v/v)). Tubulin-GAL4 (BDSC #5138) was used as the driver line in all experiments unless otherwise mentioned. A TRiP-line expressing dsRNA for luciferase (BDSC #31603) was used as the Tub-GAL4 control. *grn* RNAi line BDSC #27658

was used for the majority of experiments. $w^{1118}$ (BDSC #6326) crossed with the *grn* RNAi line BDSC #27658 was used for the *grn* RNAi control. Grain RNAi line BDSC #33746 was used to confirm the phenotype specificity. The Grn-GFP (BDSC #58483) and *ACC-lacZ* (BDSC #10790) lines were used in the midgut to monitor Grain expression and activity, respectively.

## Analysis of enriched motifs

DREME [33] was used for *de novo* motif discovery for all the datasets. TOMTOM [34] was used for motif comparison and identification with the database from Nitta et. al [35] in addition to *Drosophila* TF motif databases provided by the MEME Suite [36]. The Markov background model was created for each dataset by using MEME tools, respectively. FASTA sequences of +/- 1 kB up-/downstream from TSS's for genes interest were fetched from *D. melanogaster* (v. Flybase R6.10) by using GenomicFeatures [37] package in R/Bioconductor.

## RNA sequencing and data analysis

1st instar larvae were collected 24 hours after egg laying to plates with LSD at a controlled density. Early 2nd instar larvae were transferred at 48 hours after egg laying to plates containing either LSD or HSD. After 8 hours of diet exposure, 20 larvae per sample were collected and snap-frozen in liquid nitrogen. The larvae were homogenised and total RNA was extracted with the Nucleospin RNA II kit (Macherey-Nagel) according to the manufacturer's instruction. Libraries were prepared with the TruSeq Stranded mRNA kit (Illumina), and RNA-sequencing was performed to an average depth of 20 million reads per sample with Illumina NextSeq500 technology using the 75bp cycle kit (Illumina).

Raw sequencing data quality of all datasets was assessed with FastQC (v.0.11.2) [38]. Trimmomatic (v.0.33) [39] was used for read trimming. The reads were required to be minimum of 36 bases long and scanned with 4-base sliding window with minimum quality score of 15 per base and 20 in both strands. Mapping to *D. melanogaster* genome (Flybase R6.10) was assessed with Tophat (v.2.1.0) [40]. HTSeq (v.2.7.6) [41] was used for strand-specific quantification of genes and reads below quality score 10 discarded. The differential expression analysis was performed with R/Bioconductor package limma [42]. Genes above CPM (counts per million)>1 in at least 2/3 replicates in at least one condition were kept. The Benjamini-Hochberg correction was used for adjusting p values [43].

Hypergeometric test was used as gene set enrichment method via R/Bioconductor package piano [44] using the following databases: KEGG [45], Reactome [46], Gene Ontology [47] and Wikipathways [48].

The heatmaps were generated using scaled log2CPM values of each sample group. The data scaling was performed separately for each dataset. The row-wise clustering was performed using correlation distance.

## ChIP-seq data analysis

Raw data of Grain (ENCSR909QHH), Gatad (ENCSR245UCO) and Sugarbabe (ENCSR008NYP) ChIP-seq datasets with their respective inputs were downloaded from the ENCODE portal [49]. Samples were mapped to *Drosophila melanogaster* genome (v. Flybase R6.10) with BWA (v.0.7.17) [50]. Reads below quality 5 were discarded, and maximum mismatches in seed were set to 2 with first 32 subsequences as seed. Unmapped, non-primary and alignments below mapping quality 30 were discarded along duplicates using samtools (v.1.4) [51] and picard (v.2.18.10) (Broad Institute, Cambridge USA). Fragment sizes were estimated with R/Bioconductor phantompeaktools [52,53]. Peaks below q-value 0.05 were called for primary chromosomes with MACS2 (v.2.1.0) [54]. Empirical blacklist for dm3 defined by

ENCODE [55] was converted to match dm6 with USCS's batch coordinate conversion tool [56]. Peaks in all replicates were defined as expressed and annotated to the nearest TSS from middle of the peak using *D. melanogaster* genome (v. Flybase R6.10) with R/Bioconductor's ChIPpeakAnno package [57]. The peak visualization was done by R/Bioconductor's karyoploteR package [58].

### Tissue-specific expression analysis

Tissue specific expression of all available larval genes was downloaded from FlyAtlas2 [59]. The genes were limited to those whose Cufflink test status was OK. The relative expression of each gene per tissue was calculated. The mean of the relative tissue specific expressions was calculated to selected gene sets or all genes.

### Pupariation assay for sugar intolerance

Virgin females from the Tub-GAL4 driver line were crossed with males from RNAi lines, and allowed to lay eggs overnight on apple juice plates (33.33% apple juice (v/v), 1.75% agar (w/v), 2.5% sugar (w/v) and 2.0% Nipagin (v/v)), supplemented with dry yeast. After 24 hours, 30 first instar larvae were collected to vials with 10% baker's yeast (w/v), 0.5% agarose (w/v), 2.5% Nipagin (v/v) and 0.7% propionic acid (v/v) in MilliQ, or to vials with the same diet with an additional 15% sugar (w/v). Pupariation and eclosion were followed and scored.

### Pupal volume measurements

Pupae of the indicated genotypes and diets from the pupariation assays were collected and pupal volume was measured as described previously [60].

### Polar metabolite extraction

1st instar larvae were collected 24 hours after egg laying to plates with LSD (low sugar diet, 10% yeast) at a controlled density (50 larvae per plate). Early 2nd instar larvae were transferred at 48 hours after egg laying to plates containing either LSD or HSD (high sugar diet, 10% yeast supplemented with 15% sucrose). After 24 hours of diet exposure, 20 larvae per sample were collected, washed with PBS and snap-frozen in liquid nitrogen. The metabolites were extracted with cold 80% acetonitrile buffer by manual homogenization with pellet pestles (Sigma), 12 strokes until sample was visually disrupted. Subsequently, the samples were centrifuged 13 000 rpm, 10 min at +4˚C and the supernatant was taken to further analysis. The protein contents of the supernatant were quantified with the Pierce BCA Protein Assay Kit (ThermoFisher).

### LC-MS analysis of polar metabolites

All samples were analyzed on Thermo Q Exactive Focus Quadrupole Orbitrap mass spectrometer coupled with a Thermo Dionex UltiMate 3000 HPLC system (Thermo Fisher Scientific, Inc.). The HPLC was equipped with a hydrophilic ZIC-pHILIC column (150 × 2.1 mm, 5 μm) with a ZIC-pHILIC guard column (20 × 2.1 mm, 5 μm, Merck Sequant). 5 μl of the samples was injected into the LC-MS after quality controls in randomized order having every 10th sample as blank. The separation was achieved by applying a linear solvent gradient in decreasing organic solvent (80–35%, 16 min) at 0.15 ml/min flow rate and 45˚C column oven temperature. The mobile phases were following, aqueous 200 mmol/l ammonium bicarbonate solution (pH 9.3, adjusted with 25% ammonium hydroxide), 100% acetonitrile and 100% water. The amount of the ammonium bicarbonate solution was kept at 10% throughout the run resulting in steady 20 mmol/l concentration. Metabolites were analyzed using an MS equipped with a

heated electrospray ionization (H-ESI) source using polarity switching and following setting: resolution of 70,000 at m/z of 200, the spray voltages: 3400 V for positive and 3000 V for negative mode, the sheath gas: 28 arbitrary units (AU), and the auxiliary gas: 8AU, the temperature of the vaporizer: 280°C, temperature of the ion transfer tube: 300°C. Instrument control was conducted with the Xcalibur 4.1.31.9 software (Thermo Scientific). The peaks for metabolites were confirmed using commercial standards (Merck Cambridge Isotope Laboratories & Santa Cruz Biotechnology). The final peak integration was done with the TraceFinder 4.1 SP2 software (Thermo Scientific) and for further data analysis, the peak area data were exported as an Excel file.

## Metabolomics data analysis

One sample (grn_HSD_A) was defined as an outlier and removed from the analysis, leaving 5 replicates for *grain* LSD, control LSD and control HSD, and 4 replicates for *grain* HSD. Thus, degrees of freedom were 15. Principal component analysis of the samples was performed (S3 Fig), showing clustering of the groups. Raw data was normalized to protein levels. Levene's test was used to assess homogeneity of variance. Two-way ANOVA was performed for all metabolites followed by Tukey's honestly significant difference test. Full statistical analysis tables of Levene's test, two-way ANOVA for diet, genotype and their interaction, and Tukey HSD post hoc, with p and F values, is available as S1 Data.

## Triacylglycerol analysis by coupled colorimetric assay

30 first instar larvae were collected in three replicates of each genotype into vials with experimental diets. 10 stage-matched early 3rd instar larvae per replicate were washed in PBS and snap-frozen in liquid nitrogen. Samples were homogenised manually with 10 strokes of a pellet pestle in PBS + 0.05% Tween 20 on ice. TAG was measured using the coupled colorimetric assay (Triglyceride Reagent–Sigma; 82449, Free Glycerol Reagent–Sigma; F6428) and normalized to protein levels as described previously [61]. Sample protein levels were determined with the Pierce BCA Protein Assay Kit (ThermoFisher).

## Triacylglycerol analysis by mass-spectrometry based lipidomics

Control and Grain-depleted first instar larvae were collected to plates with LSD (low sugar diet, 10% yeast) at a controlled density (50 larvae per plate). Early third instar larvae were transferred at 72 hours after egg laying to plates containing 10% yeast + 1% glucose for 24 hours. 15 larvae per sample were washed in MilliQ and transferred to pre-chilled experimental vials, where their insides were exposed by making an incision with forceps in the cuticle immediately before snap-freezing the samples in liquid nitrogen. Larvae total lipids were extracted according to the Folch procedure [62] and dissolved in chloroform/methanol 1:2. Immediately before mass spectrometry, 2% $NH_4OH$ and an internal standard mixture were added. The samples were then infused into a triple quadrupole mass spectrometer (Agilent 6410 Triple Quadrupole; Agilent Technologies, Santa Clara, CA) at a flow rate of 10 µl/min, and spectra were recorded. The obtained mass spectra were processed by MassHunter software (Agilent Technologies, Inc. California, USA). TAG species were detected as $(M + NH_4)^+$ ions, and the total concentrations of TAG species were quantified against the internal standard TAG 54:3 (Sigma, T7140 Glyceryl trioleate). Phosphatidylethanolamine (PE) species were selectively detected using a head-group specific MS/MS scanning mode for neutral loss of 141 (NL141) and quantified against internal standards PE 28:2 and PE 40:2 (both synthesized in-house). The lipid species spectral intensities were converted to concentrations and molar % compositions by using the internal standards and LIMSA software [63]. The chain length index (CLI)

for TAGs was calculated from relative molar % values using the formula [(40*Σ40CTAG)
+(42*Σ42CTAG)+(44*Σ44CTAG)+(46*Σ46CTAG)+(48*Σ48CTAG)+(50*Σ50CTAG)
+(52*Σ52CTAG)+(54*Σ54CTAG)]/100/3, where e.g. Σ40CTAG refers to the sum of the
molar % values of all the TAG species having the total of 40 carbons in their three acyl chains.
Thus, the CLI represents the average chain length of one acyl chain.

## Lipid droplet staining and quantification

Fat bodies from pre-wandering third instar larvae raised on a LSD and HSD were dissected
and fixed in 4% formaldehyde for 30 minutes. Samples were washed three times with PBS, and
stained with LipidTOX (H34477, Thermofisher) 1:400 in PBS for 30 minutes at room tempera-
ture. Fat bodies were again washed three times, and mounted in Vectashield Mounting Media
with DAPI (Mediq). Samples were imaged using Aurox Clarity LFC HS microscope, and the
images were processed with ImageJ software (NIH). For the lipid droplet analysis, the middle
slices were selected from confocal stacks of fat bodies, and the outlines of the cells were manu-
ally defined. Droplet sizes were identified by the Analyze Particles function in FIJI [64].

## RNA extraction and tissue-specific quantitative RT-PCR

First instar larvae were collected in 4 replicates into plates with LSD at a controlled density. 48
hours later early third instar larvae were transferred for 8 hours to LSD or HSD. Midguts, fat
bodies, carcasses and Malphigian tubules of 5 larvae from each replicate were dissected and
snap frozen in liquid nitrogen. 4 biological replicates per genotype and dietary treatment were
collected. Tissues were manually homogenized, and total RNA was extracted with the Nucleos-
pin RNA II kit (Macherey-Nagel) according to the manufacturer's protocol. The extracted
RNA was reverse transcribed into cDNA with the SensiFAST cDNA Synthesis kit (Bioline)
according to the manufacturer's protocol. For the Grn target gene expression analysis, the bio-
logical cDNA replicates were pooled and quantitative RT-PCR was run using the SensiFAST
SYBR No-ROX Kit (Bioline) according to the manufacturer's protocol with the Light Cycler
480 Real-Time PCR System (Roche) in three technical replicates. For *grain* expression level
measurements, the same amount of RNA was reverse transcribed into cDNA for all samples
across all tissue types, and the biological replicates were run separately in two technical
replicates.

Primer sequences:
CDK7 F: 5'-GGGTCAGTTTGCCACAGTTT-3',
CDK7 R: 5'-GATCACCTCCAGATCCGTG-3',
Cabut F: 5'-CCTCTGCGATGCCTCGCTCC-3',
Cabut R: 5'-CACCTTCGGCGGAACCCTGC-3',
Smox F: 5'-GGTCAGAGAAGGCCGTCAAG-3',
Smox R: 5'-AGTTCTGTGTGGAGATCGCG-3',
FASN1 F: 5'-CTCCACCATCGAGGAGTTCA-3',
FASN1 R: 5'-CTCCACCATCGAGGAGTTCA-3',
ACC F: 5'-GGCTATGCTGCGCTTAACA-3',
ACC R: 5'-GCCTCTGTTTTGTGGGTGAC-3',
ATPCL F: 5'-TAGCCGACGTGAAGAGCAAG-3',
ATPCL R: 5'-ACAAACTTGGCAATGCGCTC-3',
AcCoAS F: 5'-GATTTTCGAAGGCACACCAT-3',
AcCoAS R: 5'-ACTTCATGAGGGCACGAATC-3',
grain F: 5'-GATCAAGCCAAAGCGAAGAC-3',
grain R: 5'-AAAGGGTTGTGGTTGTGGTC-3',

grain F: 5'-TCGTCCACTCACCATGAAAA-3',
grain R: 5'-ACCACCGAGACCCTTCTTCT-3'.

## Immunohistochemistry and nuclear intensity measurements

First instar larvae were reared at controlled density for 48 hours on plates containing LSD. Early third instar larvae were transferred for eight hours feeding to plates containing holidic medium [65] without sugar (LSD) or supplemented with 15% sucrose (HSD). The use of holidic medium was required to eliminate the high autofluorescence of the yeast-based diets and to improve the signal-to-noise ratio in fluorescent imaging. Midguts were dissected and fixed in 4% formaldehyde overnight in +4˚C. Midguts were washed with 0.1% Triton-X 100 in PBS and blocked in 1% bovine serum albumin for 1 h. Midguts were then stained with anti-β-Galactosidase (1:1000) antibody (MP Biomedicals) overnight in +4˚C, then washed 4 times and stained with anti-rabbit alexa 568 (1:1000) (Life Technologies) for 2 hours. Midguts were again washed 4 times, and were then mounted in Vectashield Mounting Media with DAPI (Vector Laboratories). Samples were imaged using the Aurox Clarity LFC HS microscope, and the images were processed with ImageJ software (NIH) using the "Stitch" script described previously [66]. For the *ACC-lacZ* intensity analysis, the middle slices were selected from confocal stacks and the outlines of the nuclei were manually defined on the basis of the DAPI signal. The mean intensity value of the cytoplasm surrounding each nucleus was subtracted from the mean intensity value of each nucleus to eliminate background signal. 3 nuclei from 4 separate anterior midguts were analyzed per sample group.

## Supporting information

**S1 Table. Top 10 results of motif prediction of upregulated and downregulated, sugar-dependent genes.**
(TIF)

**S2 Table. Log-rank test results of pupariation kinetics of Grain depleted and control animals.**
(TIF)

**S1 Data. Relative levels and statistical data of all metabolites analysed by metabolomics.**
(XLSX)

**S2 Data. Numerical original data.**
(XLSX)

**S1 Fig. Overlap between sugar-responsive genes and GATAd targets. A.** GATA TF (GATAe, Srp, Pnr, GATAd) database matches to *de novo* predicted binding motif in 1B and their statistics (TOMTOM). **B.** Venn diagram of GATAd direct targets (ChIP-seq, ENCODE dataset ENCSR245UCO) and genes upregulated on high sugar diet (RNA-seq, adj.p.val<0.05). **C.** Venn diagram of GATAd direct targets (ChIP-seq, ENCODE dataset ENCSR245UCO) and genes downregulated on high sugar diet (RNA-seq, adj.p.val<0.05). **D.** Venn diagram of Grain direct targets (ChIP-seq, ENCODE dataset ENCSR909QHH) and genes downregulated on high sugar diet (RNA-seq, adj.p.val<0.05).
(TIF)

**S2 Fig. Sugar tolerance and pupal volumes determined using an independent Grain RNAi.**
**A.** *grain* (*grn*) knockdown (Tub-GAL4) by an alternative RNAi-line (BDSC #33746) leads to developmental delay as compared to control (BDSC #31603) on both diets, but is more pronounced on a HSD. (N = 3 for control and *grain* RNAi on LSD, 2 for *grain* RNAi on HSD, 30

per replicate). **B.** *grain* knockdown (Tub-GAL4) by an alternative RNAi-line (BDSC #33746) leads to reduced pupal volume on a HSD as compared to control (BDSC #31603). (N = 30). **C.** Relative *grain* expression following Tub-GAL4-driven knockdown by two independent RNAi lines (#1: BDSC #27658; #2: BDSC #33746), compared to controls (Tub-GAL4>TriP control BDSC #31603 and UAS-*grain* RNAi BDSC #27658). **D and E.** Knockdown of *grain* (Tub-GAL4) by two independent RNAi (**D**: BDSC #27568, **E**: BDSC #33746) leads to increased pupal lethality on both experimental diets. (N = 4 in (**D**), N = 3 for control and *grain* RNAi on LSD, 2 for Grain RNAi on HSD in (**E**)). Data information: N indicates the number of biological replicates. Error bars display standard deviation. **(A).** Log-rank test (Df = 3), p.val.<0.01. **(B):** Two-way ANOVA found a significant interaction effect between diet and genotype (F (2,194) = 3.715, p.val = 0.02611). A Tukey HSD was performed with p-values indicated on the graph as indicated below. **(C):** Two-way ANOVA found no significant interaction effect between diet and genotype. Diet (F (1,16) = 5.040, p.val = 0.0393) and genotype (F (3,16) = 17.802, p.val = 2.37e-05) effects were found to be significantly different. A Tukey HSD was performed with p-values indicated on the graph as indicated below. **(D):** Two-way ANOVA found no significant interaction effect between diet and genotype. A genotype (F (2,16) = 368.184, p.val = 4.18e-14) effect was found to be significantly different. A Tukey HSD was performed with p-values indicated on the graph as indicated below. **(E):** Two-way ANOVA found no significant interaction effect between diet and genotype. Diet (F (1,7) = 7.067, p.val = 0.0325) and genotype (F (1,7) = 316.988, p.val = 4.35e-07) effects were found to be significantly different. A Tukey HSD was performed with p-values indicated on the graph as follows: *p.val<0.05, **p.val<0.01, ***p.val<0.001. LSD: 10% yeast, HSD: 10% yeast + 15% sucrose.
(TIF)

**S3 Fig. PCA plot of metabolomics samples.**
(TIF)

**S4 Fig. Grain-dependent expression of Aldehyde dehydrogenase III. A.** Expression of *aldh-III* (RNA-seq) is strongly downregulated by *grain* knockdown (Tub-GAL4>*grain* RNAi BDSC #27568) as compared to control (Tub-GAL4>TriP control BDSC #31603) 2nd instar larvae on both a low sugar diet and after 8 hours sugar exposure. (N = 4, 20 per replicate). ***adj.p.val<0.001. LSD: 10% yeast, HSD: 10% yeast + 15% sucrose. **B.** Binding profiles of Grain in *aldh-III* promoter (ChIP-seq, ENCODE dataset ENCSR909QHH) showing that *aldh-III* is a direct target of Grain. Purple bar indicates the called peak are in Grain ChIP-seq.
(TIF)

**S5 Fig. Overlap between Grain and Mlx target genes and *sug* expression upon *grain* knockdown. A.** Venn diagram displaying the overlap between genes that are upregulated (RNA-seq) in 2nd instar larvae after 8 hours of high sugar diet in Grain- and Mlx-dependent manner (adj.p.val<0.05). **B.** Expression of *sugarbabe* (RNA-seq) is not affected by *grain* knockdown (Tub-GAL4>*grain* RNAi BDSC #27568) as compared to control (Tub-GAL4>TriP control BDSC #31603) 2nd instar larvae on both a low sugar diet and after 8 hours sugar exposure. (N = 4, 20 per replicate). CPM: counts per million.
(TIF)

**S6 Fig. Changes in TAG species and lipid droplets upon Grain depletion. A.** Composition of TAG lipid species (mol% profiles measured by mass spectrometry-based lipidomics) in Grain depleted (*grn*) (Tub-GAL4>*grain* RNAi BDSC #27658) and control (Tub-GAL4>TriP control BDSC #31603) pre-wandering 3rd instar larvae after being fed a low sugar diet with 1% glucose for 24 hours. (N = 4, 15 larvae per sample). **B.** Example of fat body lipid droplet identification by the Analyze Particles function in FIJI of Grain depleted (Tub-GAL4>*grain* RNAi

BDSC #27658) and control (Tub-GAL4>TriP control BDSC #31603) fat bodies of early 3$^{rd}$ instar larvae raised on a diet either low or high in sugar.
(TIF)

**S7 Fig. Relative tissue specific expression of *grain* and its sugar responsive target genes *vs*. all *Drosophila* genes. A.** Relative *grain* mRNA expression in larval fat body, Malpighian tubules, carcass, and midgut. Midgut samples were used as reference samples, and expression of *CDK7* was used for normalization. (N = 4, tissues from 5 larvae per sample). **B**. Tissue specific expression of differentially regulated, sugar-responsive Grain target genes (RNA-seq, adj. p.val <0.05, LFC +/-0.5). The relative expression of sugar-responsive Grain target genes across tissues was calculated based on FlyAtlas2 expression data. **C**. Tissue specific relative expression of all *Drosophila* genes present in FlyAtlas2. The relative expression of selected genes across tissues was calculated based on FlyAtlas2 data. Data information: N indicates the number of biological replicates. Error bars display standard deviation. **(A)**. One-way ANOVA found a significant interaction effect between tissues (F (3,12) = 11628, p.val < 2e-16). A Tukey HSD was performed with p-values indicated on the graph as follows: *p.val<0.05, **p.val<0.01, ***p.val<0.001.
(TIF)

## Acknowledgments

We thank Heini Lassila, Juhana Juutila, Josef Gullmets, and Heikki Räikkönen for technical help. Merja Heinäniemi and Jarkko Salojärvi are thanked for expert advice on bioinformatics. Anas Kamleh, Gillian Mackay, and Karen Vousden are acknowledged for their support on metabolomics set-up. Imaging was supported by the Light Microscopy Unit of the Institute of Biotechnology, *Drosophila* work was supported by the Hi-Fly core facility, RNA sequencing was performed by DNA Sequencing and Genomics of the Institute of Biotechnology, all supported by Biocenter Finland and Helsinki Institute of Life Science.

## Author Contributions

**Conceptualization:** Ville Hietakangas.

**Formal analysis:** Krista Kokki, Nicole Lamichane, Anni I. Nieminen, Hanna Ruhanen, Jack Morikka.

**Funding acquisition:** Nicole Lamichane, Ville Hietakangas.

**Investigation:** Krista Kokki, Nicole Lamichane, Hanna Ruhanen, Bohdana M. Rovenko, Essi Havula.

**Methodology:** Krista Kokki, Anni I. Nieminen, Marius Robciuc, Reijo Käkelä.

**Project administration:** Ville Hietakangas.

**Supervision:** Ville Hietakangas.

**Visualization:** Krista Kokki.

**Writing – original draft:** Krista Kokki, Nicole Lamichane, Anni I. Nieminen, Jack Morikka, Reijo Käkelä, Ville Hietakangas.

**Writing – review & editing:** Hanna Ruhanen, Marius Robciuc, Bohdana M. Rovenko, Essi Havula, Ville Hietakangas.

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
