## [Decision Letter · Decision Letter 0]

16 Mar 2021

Dear Ville,

Thank you very much for submitting your Research Article entitled 'Metabolic gene regulation by Drosophila GATA transcription factor Grain' to PLOS Genetics.

The manuscript was fully evaluated at the editorial level and by independent peer reviewers. The reviewers appreciated the attention to an important problem, but raised some substantial concerns about the current manuscript. Based on the reviews, we will not be able to accept this version of the manuscript, but we would be willing to review a much-revised version. We cannot, of course, promise publication at that time.

If you decide to revise the manuscript for further consideration at PLOS Genetics, please aim to resubmit within the next 60 days, unless it will take extra time to address the concerns of the reviewers, in which case we would appreciate an expected resubmission date by email to plosgenetics@plos.org.

[LINK]

We are sorry that we cannot be more positive about your manuscript at this stage. Please do not hesitate to contact us if you have any concerns or questions.

Yours sincerely,

Gregory S. Barsh

Editor-in-Chief

PLOS Genetics

Gregory Copenhaver

Editor-in-Chief

PLOS Genetics

Reviewer's Responses to Questions

**Comments to the Authors:**

Reviewer #1: PGENETICS-D-21-00232

Metabolic gene regulation by Drosophila GATA transcription factor Grain

Kokki et al.

In this study Kokki et al. describe characterizing genes whose regulation by the Grain transcription factor (TFs) are affected by sugar levels. They start by re-evaluating a previously published data set identifying sugar-regulated genes in 2nd instar Drosophila larvae. In silico motif identification led to their study of the role of Grain (grn) TF. They carried out ubiquitous knockdown of grain and identified developmental delays and reduced pupal volume. They then show target genes that are downregulated in grn-RNAi, under high (HSD) and low (LSD) sugar diets. Subsequent analyses identify a number of metabolic alterations in grn-RNAi animals. The authors next probe if other TFs might act similarly and identify 66 genes that are downregulated when either sugarbabe or grain are depleted. The effects with grain were most pronounced on LSD. The authors then go on to investigate if both Sug and Gr might regulate the same genes and show overlapping binding sites in promoters of a number of genes. Finally, the authors show that lipogenesis and lipid droplet area is affected in grn depleted animals.

Overall this was a well written and thorough analysis that reveals new information about transcriptional targets of Grain under difference diet conditions. I have some minor concerns and suggestions. It seems that the information will be valuable to scientists interested in how diet contributes to changes in gene expression. I urge them to do follow up studies on the developmental phenotypes, and whether they can identify roles for any of the target genes in those processes.

Comments and questions:

1. The authors highlight the effects seen in LSD and seem to downplay that some genes are affected under HSD as well. Given that they conclude in talking about Grain and Sug, perhaps they could also focus on some HSD targets, at leats in the discussion.

2. Fig. 3 shows that a number of glycolytic genes were downregulated in grn-RNAi

conditions, either under LSD or both. They followed this up with LC-MS metabolomics. Given that some glycolytic genes were also downregulated on a HSD do the metabolomics data sets show similar effects? Why are some metabolites significantly different between HSD and LSD in grn-RNAi while others are not?

3. Fig. 3B shows a prediction of the tissues in which they believe the genes are altered, but without experimentally determining this, it seems this is not a robust data set.

4. Fig. 6 shows a comparison of the transcriptional profiles of grn-RNAi and knockdown of sugarbabe, which is another sugar-responsive TF. In Fig. 6B, the data sets for the two control groups are not totally consistent. Do the authors have any thoughts on that? Also, it would be easier to compare the effects of sugar diet if, for example, the control and experimental LSD were side by side.

5. Fig. 6D and E are reversed. Why are the data for sug not included with grn for the genes that they propose are being co-regulated?

6. The authors claim in their abstract that Grn and Sug both control lipogenic genes, yet the data is only shown for grn. In their discussion they go so far as to suggest that Sug could be a cooperative binding partner for Grain. I don’t feel their data support that statement since they show their effects on target genes are different under LSD/HSD and no other relevant data Sug effects are shown. According to their paper these two TFs regulate target genes under very different conditions. I believe they try to explain this in the discussion, but I think they also need to soften their interpretation of this data set.

Reviewer #2: In this interesting manuscript, Kokki and Lamichane et al. use a bioinformatic approach to identify GATA response elements in the promoters of genes that are co-regulated by dietary sugar in Drosophila. They find that the identified response elements closely match DNA sequences bound by the GATA family member grain. Whole-animal loss of grain leads to reduced survival of larvae to the pupal stage, a delay in pupation, and reduced pupal volume in animals on a high sugar diet. RNA-sequencing in animals with ubiquitous grain knockdown shows reduced expression of a number of glycolytic enzymes in grain knockdown larvae on a low sugar diet. Somewhat paradoxically, levels of glycolytic intermediates and TCA cycle components are elevated in larvae with ubiquitous grain knockdown, with strongest effects occurring on low sugar. The authors also show that grain participates in a network of transcription factors that are activated in response to dietary sugar. Mlx is a positive regulator of grain and grain positively regulates expression of cabut and smox on a low sugar diet. Furthermore, grain and the transcription factor sugarbabe bind to overlapping regions on promoters of genes such as ACC, FASN1 and ATPCL. This paper adds to our understanding of nutrient sensing and signaling, and it would benefit from additional work to determine what tissues grain acts in to regulate glucose homeostasis.

Major comments.

1. Because grain mutants exhibit embryonic lethality, the authors used two different RNAi transgenes to knockdown grain ubiquitously, demonstrating effects on survival on high sugar. What is the degree of grain knockdown with these transgenes?

2. In Figure 3B, the expression pattern of genes regulated by sugar in a grain-dependent manner is assessed using a computational approach (FlyAtlas2 queries). Midgut and fat body are the two dominant tissues. What is grain expression in these tissues?

3. Is grain expression regulated by dietary sugar in a tissue-specific manner?

4. RNA-sequencing of whole larvae is used to examine gene expression changes in response to ubiquitous grain knockdown. Does grain knockdown lead to equivalent changes in putative target genes (Hex-A, Pgi, AcCoAs) in midgut and fat body?

5. The transcription factors Smox and cabut are positively regulated by grain (Figure 5). Can forced expression of either of these rescue glycolytic or lipogenic gene expression in larvae with grain knockdown?

6. Does loss of grain alter expression of sugarbabe?

7. Loss of grain leads to late larval/early pupal lethality on high sugar but not low sugar (Figure 2). However, changes in glycolysis and TCA cycle intermediates (Figures 3, 4 and 6) and gene expression occur primarily on low sugar. Might these apparent discrepancies in sugar dependency relate to distinct sets of genes that drive the transition to metamorphosis rather than metabolism? Can the relatively minor effect sizes on metabolite levels observed with loss of grain explain the high sugar diet lethality?

Minor comments.

1. Throughout the manuscript, grain is inconsistently capitalized. The FlyBase gene name is “grain”, so it should be all lowercase. The authors may want to perform a similar check on other gene names in the paper.

2. In the legend for Figure 6, the text describing panels D and E is swapped.

3. Line 160-161, a verb (“was”) is missing.

Reviewer #3: This is an interesting manuscript that identifies Grainy as a novel regulator of transcription in response to dietary sugar in Drosophila. Over the last years, the Hietakangas lab has been piecing together the components of the transcriptional network that respond to dietary nutrients using the fly as a model organism, and this manuscript is an interesting and useful addition towards generating a complete picture of how an animal responds to diet. Since Grainy is evolutionarily conserved to humans, it also raises the possibility that the human genes may have similar roles in terms of transcription and metabolism. Hence I think this manuscript will be of interest to a broad audience of people studying metabolism and transcription. The data are also of good quality.

That said, there is one major issue that I think should be addressed prior to publication. I see a disconnect between the clear metabolic consequences of Grn loss-of-function and the very mild transcriptional effects that are being reported here. I find it convincing that Grn regulates metabolism. For instance, the increases in metabolites on a low-sugar diet upon Grn loss-of-function shown in Fig 4 are clear and significant, as is the drop in survival on a high-sugar diet shown in Fig 1B and the reduced TAG levels shown in Fig 7A.

However, almost all of the transcriptional effects being shown, especially in the HSD, are extremely small in magnitude, and often not statistically significant, so that it is unlikely this would carry over to a detectable change in the corresponding protein levels. (In the HSD upon Grn loss-of-function, Hex-A in Fig 3C only drops by 5% and it’s not significant, Gapdh1 doesn’t drop, Gapdh2 doesn’t drop, PyK doesn’t drop, cbt in Fig 5 also only drops by 5% and not significantly.) If protein levels aren’t changing, this can’t explain the metabolic phenotypes. I think this issue needs to be resolved. In sum, I think it would be important to show at least one target gene that is changing at the protein level that can link the transcriptional effects to the metabolic ones.

I can think of several different reasons for this disconnect, but I can’t guess which one is correct, so I mention several of them:

1. Much of the manuscript focuses on the transcriptional changes happening in the HSD condition. I assume this is because historically the authors identified Grn by looking at the HSD gene expression data (Fig 1). However, almost all the transcriptional data, the metabolite data, and the metabolic data show almost no effect of Grn loss-of-function in the HSD. Instead, all the phenotypes are more pronounced in the LSD. The only outlier that I can see is the survival curve show in Figure 2B which shows a survival defect in the HSD and not the LSD. Given that this does not seem to reproduce with a 2nd independent RNAi (Suppl. Fig. 2A), it makes me wonder if maybe the survival effect in main Fig 2B is an off-target artefact that is a red-herring causing the authors to focus much of the storyline on HSD? I think if the phenotype shown in Fig 2B cannot be reproduced with an independent RNAi, it is advisable to remove it. The metabolic consequences of Grn loss-of-function don’t necessarily need to lead to death to be biologically meaningful.

2. All the transcriptional data show a larger magnitude effect upon Grn loss-of-function in the LSD compared to the HSD. Why is that? One option is that Grn becomes more activated (e.g. post-translationally) in the LSD condition. At first glance, this doesn’t seem to fit with the fact that some genes which look like Grn targets are increasing in the HSD compared to the LSD (e.g. Hex-A, cbt). However, all these genes increase in expression also when Grn RNAi animals are shifted from LSD to HSD (ie the 3rd to the 4th box plot in Figures 3C, 5A, 5C, etc). This means that their induction upon HSD is Grn independent and due to other transcription factors. Instead, all these target genes drop more upon Grn loss-of-function in the LSD condition, indicating that Grn is more active in the LSD. Hence maybe the focus of the manuscript in terms of linking transcriptional changes to metabolic changes should be in the LSD condition.

For instance, the authors write

“Interestingly, cabut expression was significantly downregulated upon Grain depletion in both LSD and HSD fed animals (Figure 5A).”

This is not true. On HSD it’s only 5% and not significant.

Why are Grn targets genes defined as ‘sugar up-regulated and grn-i downregulated” (e.g. Figure 6A) if it looks like Grn is rather activated on a LSD? Then the targets would be “sugar down-regulated”. Are the authors maybe looking at the wrong set of target genes? Is the Grn binding motif enriched in the set of genes that is up-regulated in LSD compared to HSD (ie the ones going down on the HSD)?

Another option is that Grn activity is not regulated by sugar, and instead it constitutively drives mild expression of these target genes. In the HSD, other transcription factors such as sug become active and mask the effect of Grn. In that case, the target genes do not need to increase in the HSD condition compared to the LSD condition.

3. The one transcriptional change that I can see in the manuscript which is large enough in magnitude that I can imagine it would lead to a change in protein levels is that of Aldh-III in the LSD, where it drops 2-fold upon Grn loss-of-function (Suppl. Fig. 4A). This might be biologically meaningful. Can Aldh-III levels be detected by western blots? Maybe all the other transcriptional effects are present but not biologically meaningful.

4. All the transcriptional changes shown in the manuscript are on whole-larval lysates, if I understand correctly. Maybe there are indeed large-magnitude changes going on in specific tissues that do not represent the bulk of the animal mass, and hence are getting diluted out when whole larvae are homogenized. Can some of these transcriptional changes be analyzed at a tissue-specific level, for instance with publicly available fluorescent transcriptional reporters (e.g. GFP transcript-traps in the genome, or synthetic constructs where genomic regions upstream of target genes were hooked up to GAL4?)

Minor Issues:

1. Fig 1B is unclear to me as currently shown. Are the binding motifs for the five drosophila GATA factors known? If so, it would be useful to show them, and hence how they differ so it becomes apparent why the grn motif matches best.

2. Fig 3C – the drop in expression of Hex-A upon grn LOF on HSD is very small in magnitude (5% drop?) and it is not even significant. It is a bit misleading as shown, with the x-axis starting at 120. This is true also for many other figures showing gene expression data. It would be more fair to show graphs with the y-axis starting at 0.

3. Fig 3C most of the genes such as Gapdh1, Gapdh2 and PyK show no drop on the HSD at all upon Grn loss-of-function. It is misleading as stated that “many glycolytic genes… were expressed at a lower level in Grain-deficient animals, either on LSD or

on both dietary conditions.” This is true for the LSD, but not for both dietary conditions.

4. Is the 30% drop in cbt upon Grn loss-of-function in the LSD shown in Fig 5A visible at the protein level?

5. The statement in the Results “Interestingly, cabut expression was significantly downregulated upon Grain depletion in both LSD and HSD fed animals (Figure 5A).” is not correct if in the HSD condition it’s dropping only 10% and not statistically significant.

6. All the changes in gene expression shown in Fig 6D are so minor in magnitude (not even a 2-fold change in any single gene), I doubt they are reflected at the protein level and hence of any biological meaning (unless there’s a large magnitude change in a specific tissue) ?

7. The citations of Fig. 6D and 5E are swapped in the Results section (page 8)

**Have all data underlying the figures and results presented in the manuscript been provided?**

Reviewer #1: Yes

Reviewer #2: Yes

Reviewer #3: Yes

PLOS authors have the option to publish the peer review history of their article (what does this mean?). If published, this will include your full peer review and any attached files.

Reviewer #1: No

Reviewer #2: No

Reviewer #3: No

---

## [Decision Letter · Decision Letter 1]

27 Sep 2021

Dear Ville,

Thank you very much for submitting your Research Article entitled 'Metabolic gene regulation by Drosophila GATA transcription factor Grain' to PLOS Genetics.

The revised manuscript was seen by the previous three reviewers. As you will see, they are all positive, and we look forward to publishing the manuscript. Reviewer #2 notes some remaining minor concerns that we ask be addressed in a final round of minor revision that should not require additional external review.

We therefore ask you to modify the manuscript according to the review recommendations. Your revisions should address the specific points made by each reviewer.

[LINK]

Yours sincerely,

Gregory S. Barsh

Editor-in-Chief

PLOS Genetics

Gregory Copenhaver

Editor-in-Chief

PLOS Genetics

Reviewer's Responses to Questions

**Comments to the Authors:**

Reviewer #1: The authors have done a very good job addressing my concerns through editing their text and the addition of several new figures and more data. I have no further concerns.

Reviewer #2: The revised article "Metabolic gene regulation by Drosophila GATA transcription factor Grain" by Kokki and Lamichane et al. has been revised, and the authors have been highly responsive to the suggestions from all reviewers. In particular, the article is significantly strengthened by new data showing tissue-specific roles for grain in the larval fat body and anterior midgut. These data clarify some points of confusion from the original manuscript regarding the impact of grain function in regulating metabolism in low versus high sugar diets.

Some edits to the text will improve the manuscript:

1. The low sugar diet is essentially a no sugar diet while the high sugar diet is 10% sucrose (lines 319-322 in the methods). This information should also be provided in the main text and/or the first figure legend to give context to the results (most readers will read the methods section last (if at all)).

2. It is possible that a diet so low in sugar leads not only to sugar deficiency but also to protein to sugar mismatch as a metabolic challenge (see for example Skorupa et al., 2008, Aging Cell). This should be considered briefly in the Discussion section.

3. Much of the data in figures 1 through 8 is from whole-animal measurements of metabolites or gene expression in control larvae or larvae with ubiquitious knockdown of grain. It would be helpful to the reader to point out in each section of the results that the data being describe are from measurements from whole larvae.

4. The authors were unable to carry out epistasis experiments with grain and Smox / cabut, so we do not yet know whether altered expression of these transcription factors upon grain knockdown underlies some of the metabolic / gene expression phenotypes described in this article. Nonetheless, this should be mentioned as a possibility in the Discussion.

Comments on figures:

1. Graphs in Figure 4B-4G lack y-axis labels.

2. For Figure 8A, if they are known, units should be added to the y-axis label. (µg TAG per mg protein, for example)

3. Please define "CLI" for Figure 8C.

Minor text edits:

1. line 236 - extra ")" symbol

2. line 281 - "a downstream of Dawdle/Smox" - is the word "target" omitted?

Reviewer #3: The authors have nicely addressed the issues raised with the initial submission. They have included a large amount of new and interesting data which make the manuscript significantly stronger. I recommend this manuscript be accepted.

**Have all data underlying the figures and results presented in the manuscript been provided?**

Reviewer #1: Yes

Reviewer #2: Yes

Reviewer #3: Yes

PLOS authors have the option to publish the peer review history of their article (what does this mean?). If published, this will include your full peer review and any attached files.

Reviewer #1: No

Reviewer #2: No

Reviewer #3: No

---

## [Editor Report · Decision Letter 2]

1 Oct 2021

Dear Dr Hietakangas,

We are pleased to inform you that your manuscript entitled "Metabolic gene regulation by Drosophila GATA transcription factor Grain" has been editorially accepted for publication in PLOS Genetics. Congratulations!

Yours sincerely,

Gregory S. Barsh

Editor-in-Chief

PLOS Genetics

Gregory Copenhaver

Editor-in-Chief

PLOS Genetics

Comments from the reviewers (if applicable):

**Data Deposition**

http://datadryad.org/submit?journalID=pgenetics&manu=PGENETICS-D-21-00232R2

**Press Queries**

---

## [Editor Report · Acceptance letter]

7 Oct 2021

PGENETICS-D-21-00232R2 

Metabolic gene regulation by Drosophila GATA transcription factor Grain 

Dear Dr Hietakangas, 

We are pleased to inform you that your manuscript entitled "Metabolic gene regulation by Drosophila GATA transcription factor Grain" has been formally accepted for publication in PLOS Genetics! Your manuscript is now with our production department and you will be notified of the publication date in due course.

With kind regards,

Anita Estes

PLOS Genetics

On behalf of:
